# Multistability and regime shifts in microbial communities explained by competition for essential nutrients

Veronika Dubinkina[1,2†], Yulia Fridman[3†], Parth Pratim Pandey[2,4†], Sergei Maslov[1,2]*

[1]Department of Bioengineering, University of Illinois at Urbana-Champaign, Urbana, United States; [2]Carl R. Woese Institute for Genomic Biology, University of Illinois at Urbana-Champaign, Urbana, United States; [3]Department of Plasma Technologies, National Research Center "Kurchatov Institute", Moscow, Russian Federation; [4]National Center for Supercomputing Applications, University of Illinois at Urbana-Champaign, Urbana, United States

**Abstract** Microbial communities routinely have several possible species compositions or community states observed for the same environmental parameters. Changes in these parameters can trigger abrupt and persistent transitions (regime shifts) between such community states. Yet little is known about the main determinants and mechanisms of multistability in microbial communities. Here, we introduce and study a consumer-resource model in which microbes compete for two types of essential nutrients each represented by multiple different metabolites. We adapt game-theoretical methods of the stable matching problem to identify all possible species compositions of such microbial communities. We then classify them by their resilience against three types of perturbations: fluctuations in nutrient supply, invasions by new species, and small changes of abundances of existing ones. We observe multistability and explore an intricate network of regime shifts between stable states in our model. Our results suggest that multistability requires microbial species to have different stoichiometries of essential nutrients. We also find that a balanced nutrient supply promotes multistability and species diversity, yet make individual community states less stable.

*For correspondence:
maslov@illinois.edu

†These authors contributed equally to this work

Competing interests: The authors declare that no competing interests exist.

## Introduction

Recent metagenomics studies revealed that microbial communities living in similar environments are often composed of rather different sets of species (*Zhou et al., 2007*; *Lahti et al., 2014*; *Lozupone et al., 2012*; *Zhou et al., 2013*; *Pagaling et al., 2017*; *Gonze et al., 2017*). It remains unclear to what extent such alternative species compositions are deterministic as opposed to being an unpredictable outcome of communities' stochastic assembly. Furthermore, changes in environmental parameters may trigger abrupt and persistent transitions between alternative species compositions (*Shade et al., 2012*; *Rocha et al., 2018*; *Scheffer and Carpenter, 2003*). Such transitions, known as ecosystem regime shifts, significantly alter the function of a microbial community and are difficult to reverse. Understanding the mechanisms and principal determinants of alternative species compositions and regime shifts is practically important. Thus, they have been extensively studied over the past several decades (*Sutherland, 1974*; *Holling, 1973*; *May, 1977*; *Tilman et al., 1997*; *Schröder et al., 2005*; *Fukami and Nakajima, 2011*; *Bush et al., 2017*).

Growth of microbial species is affected by many factors, with availability of nutrients being among the most important ones. Thus, the supply of nutrients and competition for them plays a crucial role in determining the species composition of a microbial community. The majority of modeling

**eLife digest** In nature, different species of bacteria and fungi often live together in stable microbial communities. Exactly which species are present in the group and in which proportion may vary between communities. Changes in the environment, and in particular in the availability of nutrients, can trigger abrupt, extensive, and long-lasting changes in the composition of a community: these events are known as regime shifts. For instance, when bodies of water receive large quantities of phosphorus and nitrogen, certain algae can start to multiply uncontrollably and take over other species. A given community can have different stable species compositions, but it was unclear exactly how variations in nutrients can influence regime shifts.

To examine this problem, Dubinkina, Fridman, Pandey and Maslov harnessed mathematical techniques used in game theory and economics and modeled all the possible stable compositions of a community. They could then predict which environmental conditions – in this case, the amount of specific nutrients – were necessary for each stable composition to exist. These models also showed which conditions could trigger a regime shift. Finally, how resilient the communities were to different types of perturbations – for instance, an invasion by new species or changes in nutrient supply – was examined.

The results show that if competing species require different quantities of the same nutrients, then the community can have several possible stable compositions and it is more likely to go through regime shifts. In addition, a small number of keystone species were identified which can drive regime shifts by preventing other microbes from invading the community. Ultimately, these results suggest ways to control microbial communities in our environment, for example by manipulating nutrient supplies or introducing certain species at the right time. More work is needed however to verify the predictions of the model in real communities of microbes.

approaches explicitly taking nutrients into account are based on the classic MacArthur consumer-resource model and its variants (*MacArthur and Levins, 1964*; *MacArthur, 1970*; *Huisman and Weissing, 1999*; *Tikhonov and Monasson, 2017*; *Posfai et al., 2017*; *Goldford et al., 2018*; *Goyal et al., 2018*; *Butler and O'Dwyer, 2018*). This model assumes that every species co-utilizes several substitutable nutrients of a single type (e.g. carbon sources). However, nutrients required for growth of a species exist in the form of several essential (non-substitutable) types including sources of C, N, P, Fe, etc. Real ecosystems driven by competition for multiple essential nutrients have been extensively experimentally studied (see recent papers; *Fanin et al., 2015*; *Browning et al., 2017*; *Camenzind et al., 2018* and references therein). The theoretical foundation for all existing consumer-resource models capturing this type of growth has been laid in *Tilman (1982)*, where a model with two essential resources has been introduced and studied. Future studies extended Tilman's approach to three and more essential resources, where it has been shown to sometimes result in oscillations and chaos (*Huisman and Weissing, 1999*; *Huisman and Weissing, 2001*; *Shoresh et al., 2008*). However, all the previously studied models accounted for just *a single metabolite* per each essential nutrient.

Here, we introduce and study a new consumer-resource model of a microbial community supplied with *multiple metabolites* of two essential types (e.g. C and N or N and P). This ecosystem is populated by microbes selected from a fixed pool of species. We show that our model has a very large number of possible steady states classified by their distinct species compositions. Using game-theoretical methods adapted from the well-known stable marriage (or stable matching) problem (*Gale and Shapley, 1962*; *Gusfield and Irving, 1989*), we predict all these states based only on the ranked lists of competitive abilities of individual species for each of the nutrients. We further classify these states by their dynamic stability, and whether they could be invaded by other species in our pool. We then focus our attention on a set of steady states that are both dynamically stable and resilient with respect to species invasion.

For each state, we identify its feasibility range of all possible environmental parameters (nutrient supply rates) for which all of state's species are able to survive. We further demonstrate that for a given set of nutrient supply rates, more than one state could be simultaneously feasible, thereby allowing for multistability. While the overall number of stable states in our model is exponentially

large, only very few of them can be realized for a given set of environmental conditions defined by nutrient supply rates. The principal component analysis of predicted microbial abundances in our model shows a separation between the alternative stable states reminiscent of real-life microbial ecosystems. We further explore an intricate network of regime shifts between the alternative stable states in our model triggered by changes in nutrient supply. Our results suggest that multistability requires microbial species to have different stoichiometries of two essential resources. We also find that well-balanced nutrient supply rates matching the average species' stoichiometry promote multi-stability and species diversity yet make individual community states less structurally and dynamically stable. These and other insights from our consumer-resource model may help to understand the existing data and provide guidance for future experimental studies of alternative stable states and regime shifts in microbial communities.

## Results

### Microbial community growing on two types of essential nutrients represented by multiple metabolites

Our consumer-resource model describes a microbial ecosystem colonized by microbes selected from a pool of $S$ species. Growth of each of these species could be limited by two types of essential resources, to which we refer to as 'carbon' and 'nitrogen'. In principle, these could be any pair of resources essential for life: C, N, P, Fe, etc. A generalization of this model to more than two types of essential resources (e.g. C, N and P) is straightforward. Carbon and nitrogen resources exist in the environment in the form of $K$ distinct metabolites containing carbon , and $M$ other metabolites containing nitrogen. For simplicity, we ignore the possibility of the same metabolite providing both types. We further assume that each of the $S$ species in the pool is a specialist, capable of utilizing only a single pair of nutrients, that is one metabolite containing carbon and one metabolite containing nitrogen.

We assume that for given environmental concentrations of all nutrients, a growth rate of a species $\alpha$ is limited by a single essential resource via Liebig's law of the minimum (*de Baar, 1994*):

$$g_\alpha(c,n) = \min(\lambda_\alpha^{(c)} c, \lambda_\alpha^{(n)} n) \quad .$$

(1)

Here, $c$ and $n$ are the environmental concentrations of the unique carbon and nitrogen resources consumed by this species. The coefficients $\lambda_\alpha^{(c)}$ and $\lambda_\alpha^{(n)}$ are defined as species-specific growth rates per unit of concentration of each of two resources. They quantify the competitive abilities of the species $\alpha$ for its carbon and nitrogen resources, respectively. Indeed, according to the competitive exclusion principle, if two species are limited by the same resource, the one with the larger value of $\lambda$ wins the competition. Note that according to Liebig's law, if the carbon source is in short supply so that $\lambda_\alpha^{(c)} c < \lambda_\alpha^{(n)} n$, it sets the value for this species growth rate. We refer to this situation as $c$-source *limiting* the growth of the species $\alpha$. Conversely, when $\lambda_\alpha^{(c)} c > \lambda_\alpha^{(n)} n$, the $n$-source is *limiting* the growth of this species. Thus, each species always has exactly one growth-limiting resource and one non-limiting resource.

In our model, microbes grow in a well-mixed chemostat-like environment subject to a constant dilution rate $\delta$ (see *Figure 1A* for an illustration). The dynamics of the population density, $B_\alpha$, of a microbial species $\alpha$ is then governed by:

$$\frac{dB_\alpha}{dt} = B_\alpha \left[ g_\alpha(c_i, n_j) - \delta \right] \quad ,$$

(2)

where $c_i$ and $n_j$ are the specific pair of nutrients defining the growth rate $g_\alpha$ of this species according to the Liebig's law (*Equation 1*). These nutrients are externally supplied at fixed rates $\phi_i^{(c)}$ and $\phi_j^{(n)}$ and their concentrations follow the equations:

$$\begin{aligned}
\frac{dc_i}{dt} &= \phi_i^{(c)} - \delta \cdot c_i - \sum_{\text{all } \alpha \text{ using } c_i} B_\alpha \frac{g_a(c_i, n_j)}{Y_a^{(c)}}, \\
\frac{dn_i}{dt} &= \phi_i^{(n)} - \delta \cdot n_i - \sum_{\text{all } \alpha \text{ using } n_i} B_\alpha \frac{g_a(c_i, n_j)}{Y_a^{(n)}}
\end{aligned}$$

(3)

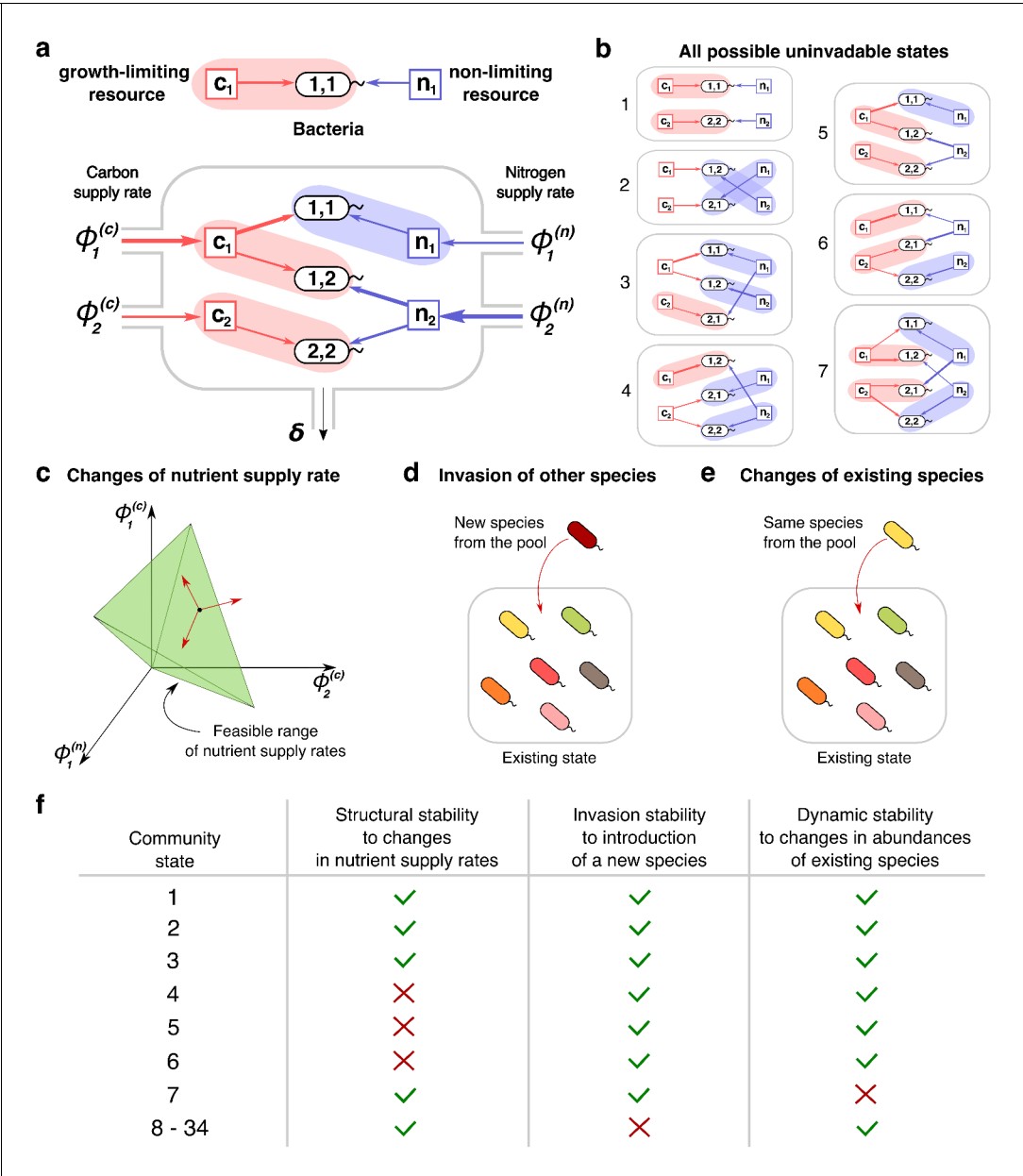

**Figure 1.** Community states and different types of their stability. (a) A schematic depiction of the proposed experimental setup and one of several possible community states in the 2C × 2N × 4S model. Several sources of carbon an nitrogen are supplied at constant rates $\phi_i^{(c)}$ and $\phi_j^{(n)}$ to a chemostat with a dilution rate $\delta$. Red and blue square nodes represent these nutrients inside the chemostat with steady state concentrations $c_1$, $c_2$ (for carbon) and $n_1$, $n_2$ (for nitrogen). They are consumed by three microbial species labeled by the pair of carbon (the first index) and nitrogen (the second index) nutrients this species consumes. Shaded ovals connect every species to its unique growth-limiting nutrient. The fourth species $B_{2,1}$ is not present in this steady state. (b) All seven uninvadable states in our realization of 2C × 2N × 4S model (see *Supplementary files 1,2* for the specific parameters). depicted using the same schematic representation as in (a). Panels (c–e) schematically illustrate three possible types of perturbations of a community state, corresponding to three different types of its stability. (c) Changes of nutrient supply rates, that may result in extinction of some of the species. Green shaded area schematically depicts the region of nutrient supply rates where a given state is feasible, red arrows represent the perturbations of nutrient supply rates. (d) Introduction of species currently absent from the system, that is invasion, that may change the set of surviving species. (e) Small fluctuations in abundances of existing species, that may disturb the dynamic equilibrium of the system and potentially drive it to another state. (f) Table that shows which stability criteria are satisfied for 34 possible states in our realization of 2C × 2N × 4S model. Note that these types of stability are in general unrelated to each other.

The online version of this article includes the following figure supplement(s) for figure 1:

**Figure supplement 1.** Distribution of volumes for stable and unstable states.

Here, $Y_\alpha^{(c)}$ and $Y_\alpha^{(n)}$ are the growth yields of the species $\alpha$ on its $c$- and $n$-resources respectively. Yields quantify the concentration of microbial cells generated per unit of concentration of each of these two consumed resources. The yield ratio $Y_\alpha^{(n)}/Y_\alpha^{(c)}$ determines the unique C:N stoichiometry of each species.

A steady state of the microbial ecosystem can be found by setting the right hand sides of *Equations 2-3* to zero and solving them for environmental concentrations of all nutrients $c_i$, and $n_j$, and abundances $B_\alpha$ of all species. We choose to label all possible steady states by the list of species present in the state and *by the growth-limiting nutrient ($c$ or $n$) for each of these species*. Thus, two identical sets of species, where at least one species is growth limited by a different nutrient are treated as two distinct states of our model. Conversely, our definition of a steady state does not take into account species' abundances. Examples of such states in a system with two carbon, two nitrogen nutrients and four species (one species for every pair of carbon and nitrogen nutrients) with specific values of species' competitive abilities $\lambda_\alpha^{(c)}$ and $\lambda_\alpha^{(n)}$ and yields $Y_\alpha^{(c)}$ and $Y_\alpha^{(n)}$ (see *Supplementary files 1,2* for their exact values) are shown in *Figure 1B*. For the sake of brevity we refer to this model as 2C × 2N × 4S.

Because each of the $S$ species in the pool could be absent from a given state, or, if present, could be limited by either its $c$- or its $n$-resource, the theoretical maximum of the number of distinct states is $3^S$ (equal to 81 in our 2C × 2N × 4S example). However, the actual number of possible steady states is considerably smaller (equal to 34 in this case). Indeed, possible steady states in our model are constrained by a variant of the competitive exclusion principle (*Gause, 1932*) (see Materials and methods for details). One of the universal consequences of this principle is that the number of species present in a steady state of any consumer-resource model cannot exceed K + M − the total number of nutrients.

We greatly simplified the task of finding all steady states in our model by the discovery of the exact correspondence between our system and a variant of the celebrated stable matching (or stable marriage) problem in game theory and economics (*Gale and Shapley, 1962*; *Gusfield and Irving, 1989*). The matching in our model connect pairs of C and N resources via microbial species using both of them. Unlike in the traditional stable marriage model, a given resource can be involved in more than one matching but cannot be limiting for more than one microbe. Thus, the competitive exclusion principle provides a number of constraints on the set of possible matchings and their stability, which are described in detail in Materials and methods and Appendix 3.

## Three criteria for stability of microbial communities

Each of the steady states identified in the previous chapter can be realized only for a certain range of nutrient supply rates. These ranges can be calculated using the steady state solutions of *Equations 2, 3*, governing the dynamics of microbial populations and nutrient concentrations, respectively (see Materials and methods). Among all formal mathematical solutions of these equations we select those, where populations of all species and all nutrient concentrations are non-negative. This imposes constraints on nutrient supply rates, thereby determining their feasible range for a given steady state (shown in green in *Figure 1C*). The volume of such feasible range has been previously used to quantify the so-called structural stability of a steady state (*Rohr et al., 2014*; *Grilli et al., 2017*; *Butler and O'Dwyer, 2018*). States with larger feasible volumes generally tend to be more resilient with respect to fluctuations in nutrient supply.

Stability of a steady state could be also disturbed by a successful invasion of a new species (see *Figure 1D*). We can test the resilience of a given state in our model with respect to such invasions. A state is called uninvadable if none of the other species from our pool can survive in the environment shaped by the existing species. *Figure 1B* shows all seven states that are uninvadable in our variant of the 2C × 2N × 4S model. Whether or not a given state is uninvadable is determined by the specific choice of parameters $\lambda_\alpha^{(c)}$, $\lambda_\alpha^{(n)}$. For example, for parameters listed in the *Supplementary file 1* the state in which $B_{12}$ is limited by carbon $c_1$, and $B_{22}$ - by carbon $c_2$ could be invaded by the species $B_{11}$. Indeed, $\lambda^{(c)}$ of $B_{11}$ is larger than that of $B_{12}$, and the nitrogen concentration $n_1$ is not limited by any species. Hence, this state is not shown in *Figure 1B*. However, the same state may turn out to be uninvadable for a different combination of parameters. The one-to-one correspondence between our model and a variant of the stable matching problem (*Gale and Shapley, 1962*) allows us to

identify all uninvadable steady states for a given choice of $\lambda_\alpha^{(c)}$, $\lambda_\alpha^{(n)}$ describing species competitiveness for resources (see Materials and methods and Appendix 3 ).

Note that, in the regime of our model, where the supply of all nutrients is high, that is $\phi^{(c,n)} \gg \delta^2/\lambda_\alpha^{(c,n)}$, invadability of individual states does not depend on supply rates. Indeed, in this regime the outcome of an attempted invasion is fully determined by the competition between species, which in turn depends only on the rank-order of competitive abilities $\lambda$ of the invading species relative to the species currently present in the ecosystem (see Materials and methods for details).

In addition to structural and invasion types of stability described above, there is also a notion of dynamic stability of a steady state actively discussed in the ecosystems literature (see e.g. *May, 1972*; *Allesina and Tang, 2012*; *Butler and O'Dwyer, 2018*). Dynamic stability can be tested by exposing a steady state to small perturbations in populations of all species present in this state (see *Figure 1E*). The state is declared dynamically stable if after any such disturbance the system ultimately returns to its initial configuration (see Materials and methods for details of the testing procedure used in our study).

We classify all the steady states in our model according to these three types of stability. The example of this classification for our realization of 2C × 2N × 4S model is summarized in *Figure 1F*. Note, that in general, one type of stability does not imply another. Out of 34 possible steady states realized for different ranges of nutrient supply rates there are only seven uninvadable ones. In the 2C × 2N × 4S model only one of the states (labelled seven in *Figure 1B*) turned out to be dynamically unstable, while for the remaining 33 states small perturbations of microbial abundances present in the state do not trigger a change of the state. Unlike two other types of stability, the structural stability has a continuous range. It could be quantified by the fraction of all possible combinations of nutrient supply rates for which a given state is feasible (referred to as state's normalized feasible range). We estimated the normalized feasible ranges of all states in the 2C × 2N × 4S model using a Monte Carlo procedure described in Materials and methods. The results are reflected in the second column of *Figure 1F*, where a structurally stable state is defined as that whose normalized feasible range exceeds 0.1 (an arbitrary threshold). In general we find that normalized feasible ranges of uninvadable states in our model have a broad log-normal distribution (see *Figure 1—figure supplement 1* for details).

It is natural to focus our attention on steady states that are simultaneously uninvadable and dynamically stable. Indeed, such states correspond to natural endpoints of the microbial community assembly process. They would persist for as long as the nutrient supply rates do not change outside of their structural stability range. Therefore, they represent the states of microbial ecosystems that are likely to be experimentally observed. From now on, we concentrate our study almost exclusively on those states and refer to them simply as stable states.

## Regime shifts between alternative stable states

The feasible ranges of nutrient supply of different stable states may or may not overlap with each other (see *Figure 2A–B* for a schematic illustration of two different scenarios). Whenever feasible ranges of two or more states overlap (see *Figure 2B*) - multistability ensues. Note that the states in the overlapping region of their feasibility ranges constitute true alternative stable states defined and studied in the ecosystems literature (*Sutherland, 1974*; *Holling, 1973*; *May, 1977*; *Fukami and Nakajima, 2011*; *Bush et al., 2017*). The existence of alternative stable states goes hand-in-hand with regime shifts manifesting themselves as large discontinuous and hysteretic changes of species abundances (*Scheffer and Carpenter, 2003*).

Every pair of states with overlapping feasibility ranges in our model corresponds to a possible regime shift between these states as illustrated in *Figure 2D* (note abrupt changes in the population $B_{11}$ at the boundary of the overlapping region). In general, a discontinuous regime shift happens in our model when one of the species ($B_{12}$ in this example) changes its growth-limiting nutrient thereby making the state invadable. It is then promptly invaded by the species present in the new state ($B_{11}$ and $B_{22}$ in our example) which may lead to immediate changes in populations of multiple species. Conversely, when feasible ranges of a pair of states do not overlap with each other but share a boundary (*Figure 2A*), the transition between these states is smooth and non-hysteretic (*Figure 2C*). It manifests itself in continuous changes in abundances of all microbial species at the boundary between states. Such continuous transitions happen in our model when the growth rate of one of

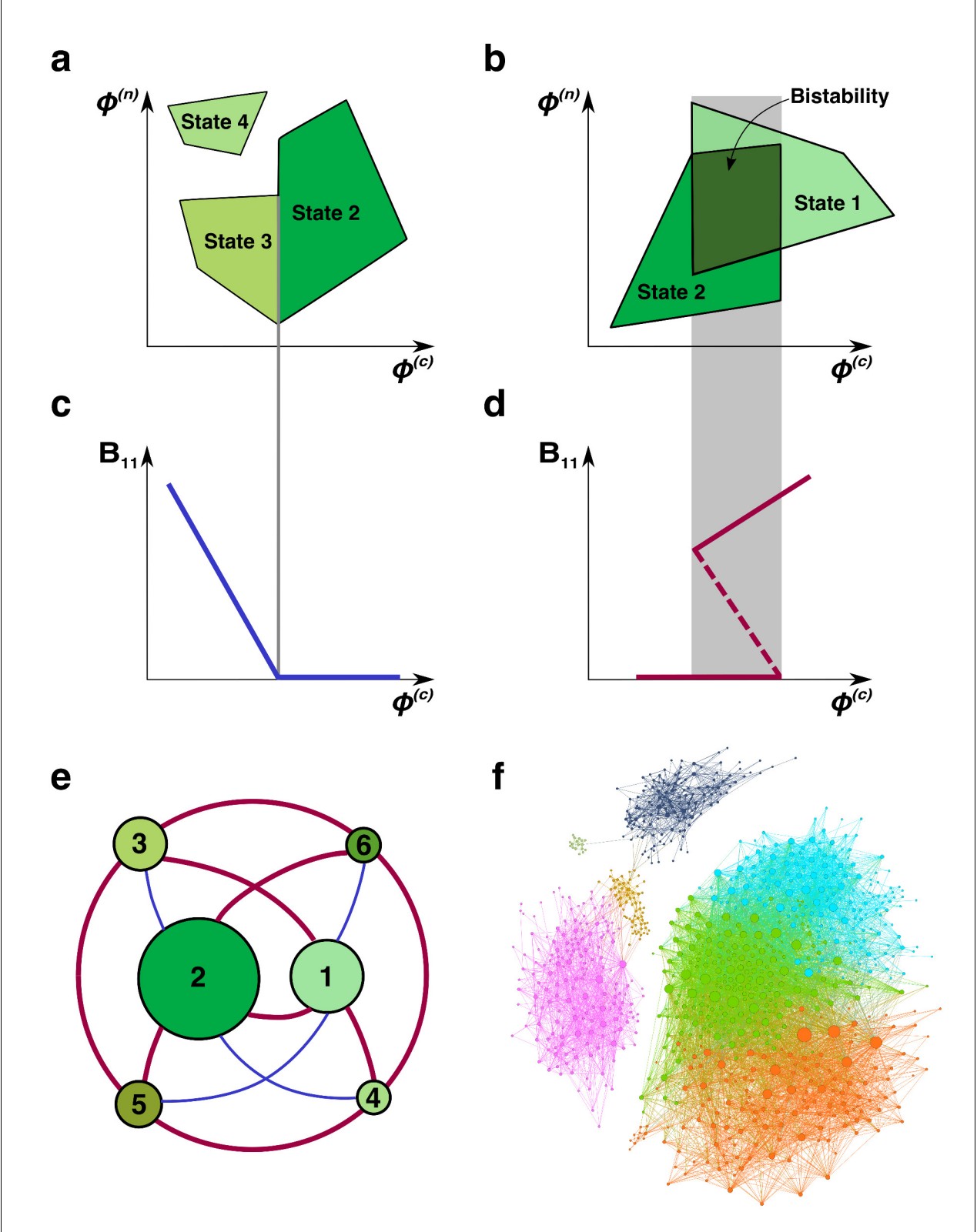

**Figure 2.** Regime shifts between alternative stable states. (a) Shaded green areas schematically depict the feasible ranges of nutrient supply rates for several stable states in our model (#2-#4 in *Figure 1B*). The feasible range of the state #4 does not overlap with that of any other state. Feasible ranges of states #2 and #3 also do not overlap but share a common boundary. Panel (b) depicts the opposite scenario of overlapping feasible ranges of another pair of stable states (#1 and #2 in *Figure 1B*). In the overlapping region (dark green), they form a pair of alternative stable states. (c) A smooth

*Figure 2 continued on next page*

*Figure 2 continued*

transition between two states at the boundary. The population $B_{11}$ of the microbial species (1,1) is plotted as a function of changing nutrient supply rate $\phi^{(c)}$ (same as the x-axis in panel (a)). Vertical gray line corresponds to the boundary between states #3 and #2. (d) A regime shift between two states. $B_{11}$ is plotted as a function of nutrient supply $\phi^{(c)}$ as it sweeps through the overlapping region (gray area) in panel (b). Note abrupt changes of $B_{11}$ at the boundaries of the overlapping region and its hysteretic behavior as expected for regime shifts. Dashed line corresponds to $B_{11}$ in a dynamically unstable state (#7 in *Figure 1B*). (e) The network of possible regime shifts between pairs of stable states in the 2C × 2N × 4S model. Each red edge represents a possible regime shift between two states it connects (overlap of their feasible ranges as in panel (d)). Each blue edge corresponds to a smooth transition between two states while changing the fluxes (as in panel (c)). Nodes correspond to six uninvadable and dynamically stable states (state labels are the same as in *Figure 1b*). Sizes of nodes reflect relative magnitudes of feasible ranges of states they represent. (f) Network of 8633 possible regime shifts between pairs of 893 uninvadable dynamically stable states in the 6C × 6N × 36S model. The size of each node reflects its degree (i.e. the total number of other stable states that a given state can shift into). The color of each node corresponds to its network modularity class calculated as described in Materials and methods.

The online version of this article includes the following figure supplement(s) for figure 2:

**Figure supplement 1.** Statistics of pairwise bistability network for 6C × 6N × 36S example.

the species ($B_{21}$ in this example) falls below the dilution rate $\delta$. This species then slowly disappears from the ecosystem thereby changing its state. When this boundary is crossed in the opposite direction, the same species ($B_{21}$) gradually appears in the ecosystem.

As expected for regime shifts, dynamically unstable states always accompany multistable regions in our model (*Scheffer and Carpenter, 2003*) (see below for a detailed discussion of the interplay between multistability and dynamically unstable states). We observed that dynamically unstable state #7 in our 2C × 2N × 4S is feasible in the overlapping region between states #1 and #2 in *Figure 2B*. The population $B_{11}$ in this state is shown as dashed line in *Figure 2D*.

We identified all possible regime shifts in the 2C × 2N × 4S model by systematically looking for overlaps between the feasible ranges of nutrient supply of all six uninvadable dynamically stable states. These regime shifts can be represented as a network in which nodes correspond to community's stable states and edges connect states with partially overlapping feasible ranges (see thick red edges in *Figure 2E*). One can see that regime shifts are possible only for of nine pairs of uninvadable states. We performed additional simulations (see Materials and mthods) looking for shared boundaries (continuous transitions) between uninvadable states and identified additional four pairs of states bordering each other (thin blue edges in *Figure 2E*). The pairs of states #5 - #6 and #3 - #4 do not directly transition to each other either continuously or discontinuously. This indicates that their feasible ranges are too far apart from each other, so that they do not have any overlaps or common boundaries.

Combining the information in *Figure 1B* and *Figure 2E* one can find that all states connected by a discontinuous regime shift in our 2C × 2N × 4S model have two distinct sets of keystone species: $B_{11}$-$B_{22}$ in one state and $B_{12}$-$B_{21}$ in another. This is because all regime shifts are driven by the same bistable switch in which these pairs of species compete and mutually exclude each other. The dynamically unstable state #7 is formed by the union of all four keystone species and, when perturbed, collapses into a state with either one or another keystone set. Conversely, states connected by a continuous transition share the same pair of keystone species. One of the 'satellite' species, that is species distinct from the keystone, gradually goes extinct when the boundary between these states is crossed. When the nutrient supply is changed in the opposite direction this species gradually invades the system.

*Figure 2F* shows a much larger network of 8633 regime shifts between 893 uninvadable dynamically stable states in the 6C × 6N × 36S realization of our model. In this model the microbial community is supplied with six carbon and six nitrogen nutrients and colonized from a pool of 36 microbial species (one for each pair of C and N nutrients) (see *Supplementary files 3*, *4*, *5*, *6* for the values of $\lambda$'s and yields). For simplicity, we did not show the remaining 165 uninvadable stable states that have no possible regimes shifts to any other states. The size of a node is proportional to its degree (i.e. the total number of other states it overlaps with) ranging between 1 and 164 with average around 20 (the degree distribution is shown in *Figure 2—figure supplement 1*).

The network modularity analysis (see Materials and methods for details) revealed seven network modules indicating that pairs of states that could possibly undergo a regime shift are clustered together in the multi-dimensional space of nutrient supply rates. This modular structure suggests the

existence of distinct sets of keystone species driving regime shifts within each module. However, the complexity of the 6C × 6N × 36S model does not allow a straightforward identification of these drivers (paired sets of keystone species and dynamically unstable states).

## Patterns of multistability

In a general case, the number of stable states that are simultaneously feasible for a given set of nutrient supply rates can be more than two. Furthermore, as the number of nutrients increases, the multistability with more than two stable states becomes progressively more common. In *Figure 3A*, we quantify the frequency of multistability with $V$ stable states occur in our 6C × 6N × 36S model across all possible nutrient supply rates (see Materials and methods for details of how this was estimated). $V - 1$ approximately follows a Poisson distribution (dashed line in *Figure 3A*) with $\lambda = 0.063$. Note that for some supply rates up to five stable states can be simultaneously feasible. However, the probability to encounter such cases is exponentially small.

We further explored the factors that determine whether multistability is possible in resource-limited microbial communities. Like in a simple special case of regime shift between two microbial species studied in *Tilman (1982)*, multistability in our model is only possible if individual microbial species have different C:N stoichiometry. This stoichiometry is given by the ratio of species' nitrogen and carbon yields. Our numerical simulations and mathematical arguments show that when all species have exactly the same stoichiometry $Y_\alpha^{(n)}/Y_\alpha^{(c)}$, there is no multistability or dynamical instability in

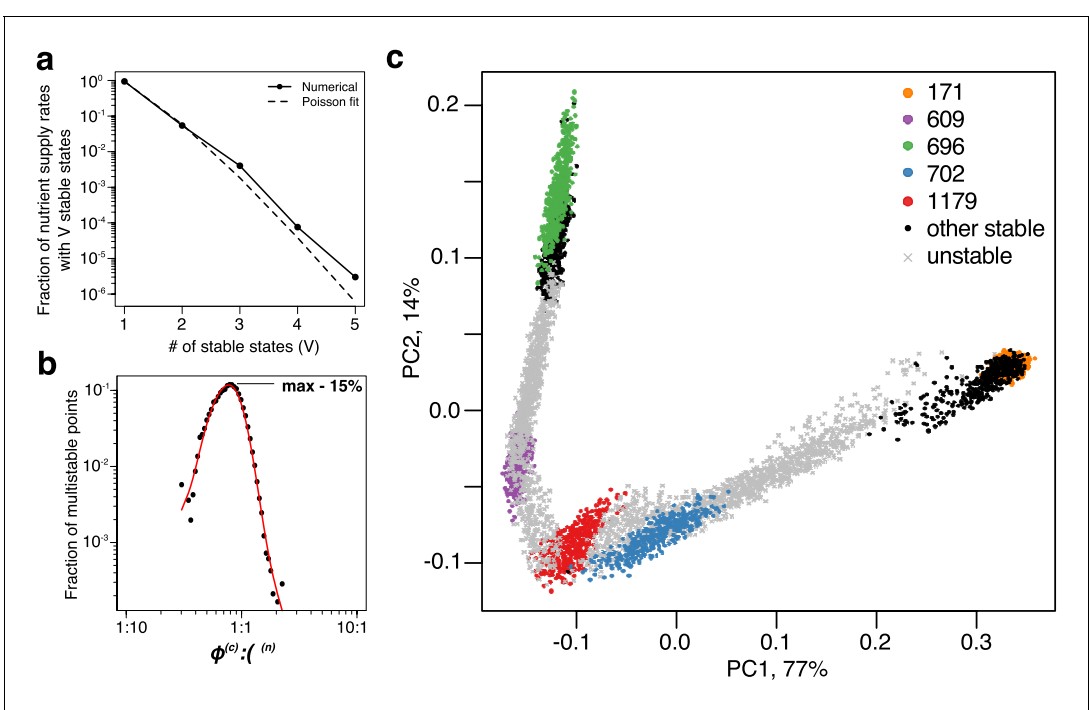

**Figure 3.** Patterns of multistability. (a) The distribution of the number, $V$, of multistable states across the entire space of nutrient supply rates. The data is based on Monte Carlo sampling of 1 million different environments (combinations of nutrient supply rates) in the 6C × 6N × 36S model. Solid circles show the fraction of all sampled environments for which $V = 1, 2, 3, 4, 5$ uninvadable dynamically stable states are simultaneously feasible. The dashed line is the fit to the data with a Poisson distribution for $v - 1$ extra states giving rise to multistability. (b) Fraction of multistable cases for different ratios of supply of two essential nutrients. The peak of the distribution is close to the balanced supply ($\phi^{(c)} : \phi^{(n)} \simeq 1 : 1$). (c) The PCA plot of relative microbial abundances in the vicinity of the environment, where $V = 5$ stable states coexist. Supply rates were randomly sampled within ±10% from the initial environment. Each point shows the first (x-axis) and the second (y-axis) principal components of microbial abundances in every uninvadable state feasible for this combination of supply rates. Colored circles label the original five stable states, black circles - several other stable states, which became feasible for nearby supply rates, and grey crosses - dynamically unstable states feasible in this region of nutrient supply rates.

The online version of this article includes the following figure supplement(s) for figure 3:

**Figure supplement 1.** The PCA plot of fractional microbial abundances.

our model (see Appendix 5 ) . That is to say, in this case for every set of nutrient supply rates the community has a unique uninvadable state, and all these states are dynamically stable.

A complementary question is whether multistable states are more common around particular ratios of carbon and nitrogen supply rates. *Figure 3B* shows this to be the case: the likelihood of multistability has a sharp peak around the well-balanced C:N nutrient supply rates. In this region multiple stable states are present for roughly 15% of nutrient supply rate combinations. Note that the average C:N stoichiometry of species in our model is assumed to be 1:1. In case of an arbitrary C:N stoichiometry, by redefining the units of nutrient concentrations and supply rates one can transform any ecosystem to have 1:1 nutrient ratio. Hence, in general, we predict that the highest chance to observe multistability will be when the ratio of nutrient supply rates is close to the average C:N stoichiometry of species in the community.

To illustrate how multistable states manifest themselves in a commonly performed Principal Component Analysis (PCA) of species' relative abundances, we picked the environment with $V = 5$ simultaneously feasible stable states in our 6C $\times$ 6N $\times$ 36S model. In natural environments, nutrient supply usually fluctuates both in time and space. To simulate this we sampled a ±10% range of nutrient supply rates around this chosen environment (see Materials and methods) and calculated species' relative abundances in each of the uninvadable states feasible for a given nutrient supply. To better understand the relationship between dynamically stable and unstable states we included the latter in our analysis. *Figure 3C* shows the first vs the second principal components of relative microbial abundances sampled in this fluctuating environment. (two more examples calculated for different multistable neighborhoods are shown in *Figure 3—figure supplement 1A–B*). One can see five distinct clusters, each corresponding to a single dynamically stable uninvadable state. Interestingly, in the PCA plot these states are separated by $V - 1 = 4$ dynamically unstable ones. Furthermore, all states are aligned along a quasi-1D manifold with an alternating order of stable and unstable states. It is tempting to conjecture that some variant of our model may explain similar arrangements of clusters of microbial abundances, commonly seen in PCA plots of real ecosystems. If this is the case, the gaps between neighboring clusters would correspond to dynamically unstable states of the ecosystem, which may be experimentally observable as long transients in community composition.

## Patterns of diversity and structural stability of states

Above we demonstrated that multistable states are much more common for balanced nutrient supply rates, that is to say, when the average ratio of carbon and nitrogen supply rates matches the average C:N stoichiometry of species in the community (see *Figure 3B*). Interestingly, a balanced supply of nutrients also promotes species diversity. In *Figure 4A*, we plot the average number of species in a stable state, referred to as species richness, as a function of the average balance between carbon and nitrogen supplies for 6C $\times$ 6N $\times$ 36S model. The species richness is the largest (around 10.5) for balanced nutrient supply rates, while dropping down to the absolute minimal value of six in two extreme cases of very large imbalance of supply rates, where the nutrient supplied in excess becomes irrelevant in competition. In this case, only six species that are teh top competitors for carbon metabolites (if nitrogen supply is plentiful) or, respectively nitrogen metabolites (if carbon is large) survive, while the rest of less competitive species are never present in uninvadable states. The number of distinct community states also has a sharp peak at balanced nutrient supply (see 3-orders of magnitude difference in *Figure 4—figure supplement 1*).

For balanced nutrient supply rates the relationship between species' competitiveness and its prevalence in the community is much less pronounced than for imbalanced ones. It is shown in *Figure 4B*, where we plot the prevalence of the species as a function of its average competitiveness. Here, the *average competitiveness rank* of a species is defined as the mean of its ranks of competitive abilities ($\lambda$ parameters of the model) for its carbon and nitrogen resources. The rank 1 being assigned to the most competitive species for a given resource (the species with the largest value of $\lambda$), while the rank 6 - to the least competitive species for this resource. Species *prevalence* is given by the fraction of all environments where it can survive. Note that all 36 species in our pool are present in some of the environments.

In general, more competitive species tend to survive in a larger subset of environments (see the dashed curve in *Figure 4B*). For example, in our pool there is one species which happens to be the most competitive for both its carbon and nitrogen sources. This species is present in all of the states in every environment. However, we also find that some of the least competitive species (those at the

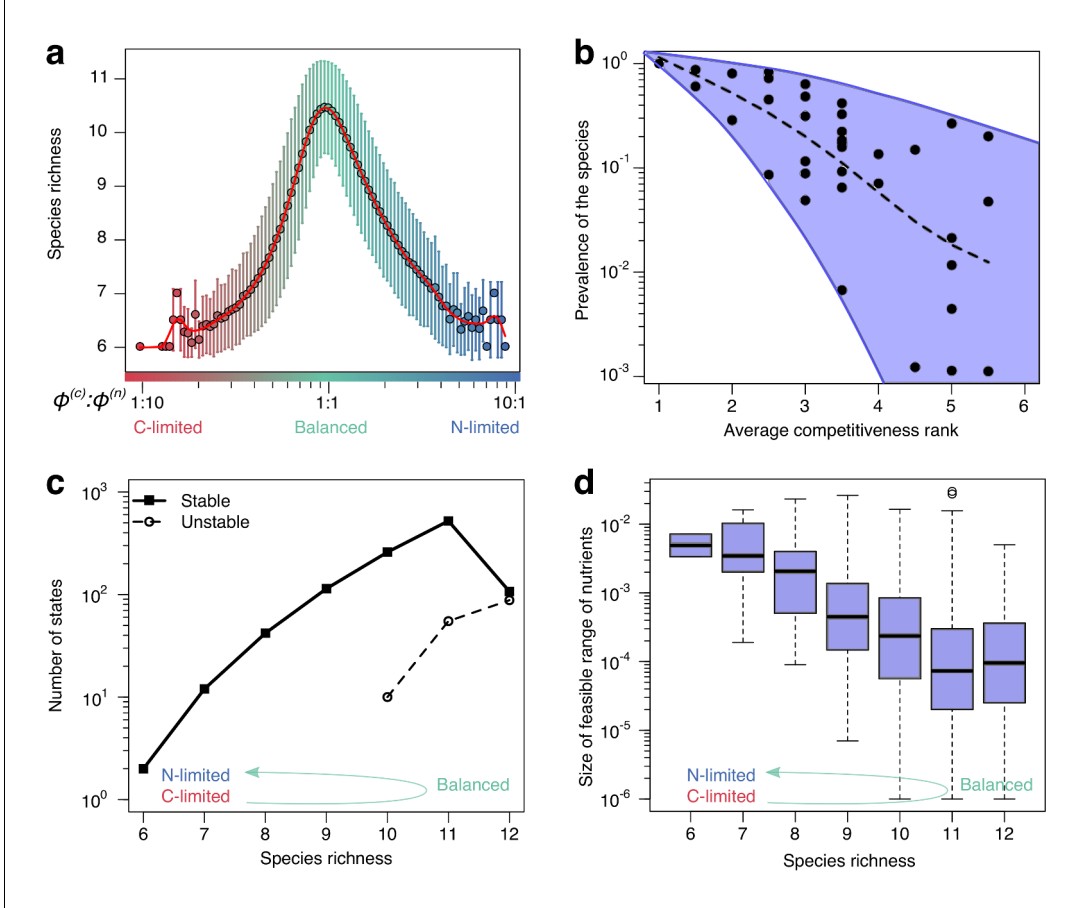

**Figure 4.** Patterns of diversity and structural stability of states. (**a**) Average species richness (y-axis) of uninvadable stable states feasible for a given $\phi^{(c)} : \phi^{(n)}$ nutrient supply ratio (x-axis). Error bars correspond to standard deviation of species richness of individual states feasible for a given nutrient supply ratio (see **Figure 4—figure supplement 1** for the number of states contributing to each point). (**b**): Scatter plot of the prevalence (y-axis) of each of the 36 species in the 6C × 6N × 36S model plotted vs its average competitiveness rank for its carbon and nitrogen sources. The latter is calculated from the rank order of $\lambda^{(c)}$ and $\lambda^{(n)}$ among all species consuming each resource (rank one corresponds to the largest $\lambda$ for this resource among all species). Species prevalence is quantified as the fraction of environments where a given species can survive. The dashed line shows the average trend. (**c**) The number of uninvadable dynamically stable (solid line) and unstable (dashed line) states with a particular species richness (x-axis). (**d**) Boxplot of nutrient feasibility ranges of uninvadable stable states plotted as a function of their species richness. All plots were calculated for the 6C × 6N × 36S model.

The online version of this article includes the following figure supplement(s) for figure 4:

**Figure supplement 1.** Number of unique uninvadable stable states that are feasible for a given $\phi^{(c)} : \phi^{(n)}$ nutrient supply ratio (x-axis is binned as in **Figure 4A**).

right end of the x-axis in **Figure 4B**) survive in a broad range of environments. For example, one species with average competitiveness rank of 5.5 corresponding to the last and next to last rank for its two resources still has relatively high prevalence of around 20%. This illustrates complex ways in which relative competitiveness of all species in the pool shapes their prevalence in a broad range of environments.

We also explore the relationship between species richness of a state (i.e. its total number of surviving species) and its other properties. **Figure 4C** shows an exponential increase of the number of uninvadable states as a function of species richness. In our 6C × 6N × 36S model all uninvadable states with less than 10 species are dynamically stable (solid line in **Figure 4C**), while those with 10 or more species can be both stable or unstable (dashed line in **Figure 4C**). Overall, the fraction of stable states to dynamically unstable ones decreases with species richness. In other words, the

probability for a state to be dynamically unstable increases with the number of species. In this aspect, our model behaves similar to the gLV model in Robert May's study (*May, 1972*).

In *Figure 4D*, we show a negative correlation between the species richness of a stable state and its feasible range of nutrient supplies. Thus in our model the number of species in an ecosystem has detrimental effect on the structural stability of the community quantifying its robustness to fluctuating nutrient supply (*Rohr et al., 2014*). The empirically observed exponential decay of state's feasible range with its number of species is well described by a two-fold decrease per each species added (see *Serván et al., 2018* and *Grilli et al., 2017* for related results in the gLV model). Note that the observed decrease in feasible range with species richness goes hand-in-hand with an increase in the overall number of states. Thus, in well-balanced environments a large number of states are carving all possible combinations of nutrient supply into many small and overlapping ranges.

Overall, the results of our model with a large number of nutrients suggest the following picture. In nutrient-balanced environments, we expect to observe a high diversity of species in the existing communities. These species can form a very large number of possible combinations (uninvadable states). Each of these states could be realized only for a narrow range of nutrient supply rates indicating their low structural stability. Moreover, in such environments we predict common appearance of multistability between some of these states.

## Discussion

The inspiration for our model was the common appearance of alternative stable states in ecosystems in general, and microbial communities in particular (*Sutherland, 1974*; *Tilman et al., 1997*; *Schröder et al., 2005*; *Fukami and Nakajima, 2011*; *Bush et al., 2017*; *Pagaling et al., 2017*; *Gonze et al., 2017*). For example, eutrophication of shallow lakes caused by algal competition for N and P is one of the best studied examples of alternative stable states and regime shifts (*Scheffer and Jeppesen, 2007*). To the best of our knowledge our model is the first consumer-resource model capable of multistability between several states, each characterized by a high diversity of species. We extend Tilman's scenario (*Tilman, 1982*) in which the growth of two species is limited by a pair of essential resources to the case of multiple nutrients of each type. This allows us to assemble complex communities with large number of co-existing species and provides additional insights into patterns of multistability in such communities.

### Multistability requires diverse species stoichiometry

We find that multistability in our model requires a mix of species with different nutrient stoichiometries. In this aspect it is similar to both the Tilman model (*Tilman, 1982*), and the MacArthur model (*MacArthur and Levins, 1964*; *MacArthur, 1970*; *Chesson, 1990*). Common variants of the MacArthur model assume identical biomass yields of different species growing on a given nutrient (*Tikhonov and Monasson, 2017*; *Posfai et al., 2017*; *Goldford et al., 2018*; *Goyal et al., 2018*; *Butler and O'Dwyer, 2018*). In this case, the absence of multistability is guaranteed by a convex Lyapunov function (*MacArthur, 1970*) guiding any dynamical trajectory of the system to its unique minimum. However, a MacArthur model with different nutrient yields of different species is capable of multistability. For some growth functions $g(C,N)$ multistability is possible even in a community of species with identical nutrient yields/stoichiometries. For example, the growth function $g(C,N) = \lambda \cdot C \cdot N$ has been numerically studied in the context of autocatalytic polymer growth and shown to be capable of multistability (*Tkachenko and Maslov, 2018*). This model had 1:1 stoichiometry: a ligation always eliminates one left end and one right end of two polymer chains and generates one autocatalytic polymer segment. Another type of growth function with two essential resources has been shown to have bistable solutions even for identical species stoichiometries (see Figure 5C in *Marsland et al., 2019b*). The Minimum Environmental Perturbation Principle introduced in this study may provide additional insights on the necessary conditions for multistability in consumer resource models.

Given that multistability in our model is impossible in communities of species with identical stoichiometries, it is reasonable to expect that the larger is the variation of C:N ratio of individual microbes, the higher is the likelihood to observe multistability. We investigated this question in the 2C $\times$ 2N $\times$ 4S model and summarized the results in *Figure 5*. It shows that the likelihood of finding

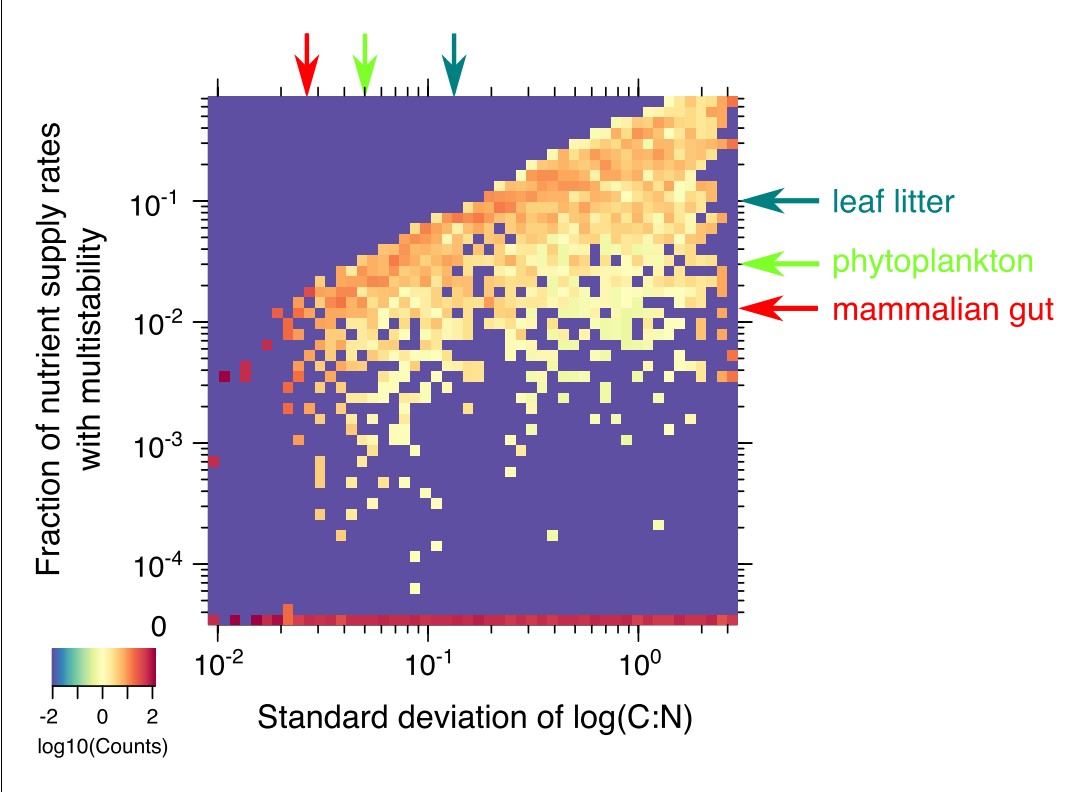

**Figure 5.** Statistics of multistability for different combinations of yields. Heatmap of the fraction of nutrient supply rate combinations that permit multistability for the 2C × 2N × 4S model with the same set of $\lambda$'s but different combinations of microbial growth yields (4000 model variants in total). Standard deviations of yields (x-axis) and the fraction of nutrient supply rates with multistability (y-axis) were logarithmically binned into 50 bins along each axis. The color scale represents log10 of the normalized count in each bin. The counts were normalized to add up to 100% in each column (same bin of the x-axis) to approximate the probability distribution of multistability fraction for given standard deviation of species stoichiometry. The bottom row (0) corresponds to 2069 yields combinations where no multistability was observed for any nutrient supply combinations in our Monte-Carlo simulations of $10^6$ flux points. The red arrow on the x-axis corresponds to the approximate yield variation for mammalian gut microbes from *Reese et al. (2018)*. The red arrow on y-axis highlights the predicted likelihood of multistability in our model for a microbial community with the same yield variation in our model. Light green arrows show an estimation of these numbers for phytoplankton species from *Finkel et al. (2016)*. Dark green arrows correspond to the soil microbes studied in *Mouginot et al. (2014)*.

nutrient supply rates with multistability systematically decreases with standard deviation of the logarithm of species stoichiometry. We also found that about half of the combinations of species stoichiometries yielded no multistable states at all. Multistability in our model is caused by a complex interplay between species' competitiveness abilities $\lambda$ and their C:N stoichiometries $\frac{Y^{(n)}}{Y^{(c)}}$. Appendix 4 explains why multistability is impossible for half of yield combinations for which the enumerator in *Equation S11* exceeds its denominator.

Nutrient stoichiometry of phytoplankton species in marine ecosystems has been known to be relatively universal with C:N:P $\simeq$ 106:16:1 known as Redfield ratio (*Redfield, 1958*). Thus species-to-species variability of C:N ratio for phytoplankton is rather small with logarithmic standard deviation estimated to be around 0.05 based on data from *Finkel et al. (2016)*. In this limit, the multistability in our model is rather unlikely (observed in ~3% of nutrient supply combinations, see green arrow in *Figure 5*). The likelihood of multistability is also low (~1%) for the mammalian gut microbiome, where variability of the logarithm of C:N ratio in different 'keystone' gut species studied in *Reese et al. (2018)* is around 0.03. The chances of multistability increase in terrestrial ecosystems such as soil, where significant deviations from the Redfield ratio have been reported (*Cleveland and Liptzin, 2007*). For example, using the data for the microbial species from grassland leaf litter community reported in *Mouginot et al. (2014)*, with log(C:N) variability of 0.12 we predict the likelihood of multistability to be around 10%.

## Multistability requires balanced nutrient supply matching the average species stoichiometry

Another important factor favoring multistability in our model is the balanced supply of two essential nutrients (see *Figure 3B*). It occurs when the average ratio of supply rates of two essential nutrients matches the average C:N stoichiometry of community's species (see *Figure 3B*). When nutrient supplies are balanced, microbial community multistability is relatively common. Furthermore, for balanced nutrients the community can be in one of many different states, characterized by different combinations of limiting nutrients. These states tend to have high species diversity (*Figure 4A*) – a trend consistent with lake ecosystems in *Interlandi and Kilham (2001)* – and relatively small range of feasible supply rates (*Figure 4D*). Hence, regime shifts can be easily triggered by changes in nutrient supply. The balanced region is characterized by a complex relationship between species competitiveness and survival, so that even relatively poor competitors could occasionally have high prevalence (species in the upper right corner of *Figure 4B*).

In the opposite limit, the supply of nutrients of one type (say nitrogen) greatly exceeds that of another type (say carbon). For such imbalanced supply, the community has a unique uninvadable state, where every carbon nutrient supports the growth of the single most competitive species. Nitrogen nutrients are not limiting the growth of any species and thus have no impact on species survival and community diversity. As a consequence, the average diversity of microbial communities in such nutrient-imbalanced environments is low (about one half of that for balanced supply conditions). This is in agreement with many experimental studies showing that addition of high quantities of one essential nutrient (e.g. as nitrogen fertilizer) tends to decrease species diversity. This has been reported in numerous experimental studies cited in the chapter 'Resource richness and species diversity' of *Tilman (1982)* as well as in recent experiments in microbial communities (*Mello et al., 2016*).

## Multistability and the total number of states are affected by tradeoffs

Species in our model are characterized by their competitiveness abilities $\lambda^{(c)}$, $\lambda^{(n)}$ and nutrient yields $Y^{(c)}$ and $Y^{(n)}$. As we showed above, the rank order of the former fully defines the total number of stable states and their invadability. On the other hand, multistability highly depends on combination of species' nutrient yields. While in the current version of our model we did not assume any specific correlations between these parameters, imposing such correlations due to various biologically motivated tradeoffs may affect multistability and the total number of states of the ecosystem.

One possibility is a negative correlation between the competitive abilities of a given species for different nutrients. Such tradeoff may exist due to a limited amount of internal resources (such as the overall number of transporters) this species can allocate for consumption of all nutrients. This type of tradeoff was shown to result in an increased species diversity in well-balanced environments, but does not lead to multistability (*Posfai et al., 2017*; *Tikhonov, 2016*). Similar negative (positive) correlations to increase (decrease) the number of stable states in a very different consumer-resource model based on the stable marriage problem (*Goyal et al., 2018*). We expect these results to also apply to our model with tradeoff between $\lambda_\alpha^{(c)}$ and $\lambda_\alpha^{(n)}$. Negative correlations between species' competitive abilities for carbon and nitrogen are expected to increase the total number of stable states in our model, while positive correlations - to decrease it.

Another possibility is a negative correlation between species' competitive ability and its yield for the same nutrient. It is known as a 'growth-yield tradeoff', which states that microbial species with faster growth on a given nutrient tend to use it less efficiently (have a smaller yield) (*Pfeiffer et al., 2001*; *Beardmore et al., 2011*; *Novak et al., 2006*). Growth-yield tradeoff is expected to increase the likelihood of multistability in our model. It could be demonstrated already in the model of *Tilman (1982)* with two species competing for two essential resources. If the species with the higher growth rate on, say, carbon source has a smaller yield on this resource than the other species - bistability always ensues. Note that, while growth-yield tradeoff is known to be common among microorganisms, the macroscopic (e.g. plant) ecosystems, which are the main focus of *Tilman (1982)*, have the opposite correlation in which species' yield $Y$ is proportional to its competitive ability $\lambda$. This type of tradeoff leads to a relative scarcity of multistability in macroscopic ecosystems.

Conversely, multistability is expected to be more common in microbial ecosystems due to the growth-yield tradeoff.

## Interplay between diversity and stability in ecosystems with multiple essential nutrients

Ever since Robert May's provocative question 'Will a large complex system be stable?' (*May, 1972*) the focus of many theoretical ecology studies has been on investigating the interplay between dynamic stability and species diversity in real and model ecosystems (*Ives and Carpenter, 2007*). May's prediction that ecosystems with large number of species tend to be dynamically unstable needs to be reconciled with the fact that we are surrounded by complex and diverse ecosystems that are apparently stable. Thus, it is important to understand the factors affecting stability of ecosystems in general and microbial ecosystems in particular .

Here, we explored the interplay between diversity and stability in a particular type of microbial ecosystems with multiple essential nutrients. We discussed three criteria for stability of microbial communities shaped by the competition for nutrients: (i) how stable is the species composition of a community to fluctuations in nutrient supply rates; (ii) the extent of community's resilience to species invasions; and (iii) its dynamical stability to small stochastic changes in abundances of existing species. Naturally-occurring microbial communities may or may not be stable according to either one of these three criteria (*Ives and Carpenter, 2007*). The degree of importance of each single criterion is determined by multiple factors such as how constant are nutrient supply rates in time and space and how frequently new microbial species migrate to the ecosystem.

Our model provides the following insights into how these three criteria are connected to each other. First, as evident from *Figure 1F*, the three types of stability are largely independent from each other. Second, communities growing on a well balanced mix of nutrients tend to have high species diversity (see peak in *Figure 4A*). The similar effect was demonstrated in other consumer resource models (*Posfai et al., 2017*; *Tikhonov and Monasson, 2017*; *Taillefumier et al., 2017*; *Marsland et al., 2019a*). However, each of the community states in this regime tends to have a low structural stability with respect to nutrient fluctuations. In environments with highly variable nutrient supplies the community will frequently shift between these states. That is to say, some of the species will repeatedly go locally extinct and the vacated niches will be repopulated by others. Furthermore, many of the steady states in this regime are dynamically unstable giving rise to multistability and regime shifts. In this sense our model follows the general trend reported in *May (1972)*. Conversely, microbial communities growing on an imbalanced mix of essential nutrients have relatively low diversity (*Figure 4A*) but are characterized by a high degree of structural and dynamic stability (see *Figure 4D* and *Figure 4C* respectively). We expect these trends to apply to a broad variety of consumer-resource models.

The existence of dynamically unstable states always goes hand in hand with multistability (*Scheffer and Carpenter, 2003*) (see the dashed line in *Figure 2D* for an illustration of this effect in our model). Interestingly, in our model we always find $V - 1$ dynamically unstable states coexisting with $V$ dynamically stable ones for the same environmental parameters (see *Figure 3C* and *Figure 3—figure supplement 1* for some examples). All states (both dynamically stable and unstable) shown in *Figure 3C* are positioned along some one-dimensional curve in PCA coordinates. This arrangement hints at the possibility of a non-convex one-dimensional Lyapunov function whose $V$ minima (corresponding to stable states) are always separated by $V - 1$ maxima (unstable stable states) as dictated by the Morse theory (*Milnor, 1963*). This should be contrasted with convex multi-dimensional Lyapunov functions used in *MacArthur (1970)*, *Case and Casten (1979)* and *Chesson (1990).*

## Extensions of the model

Our model can be extended to accommodate several additional properties of real-life microbial ecosystems. First, one could include generalist species capable of using more than one nutrient of each type. The growth rate of such species is given by:

$$g_\alpha = \min\left( \sum_{i\,\text{used by}\,\alpha} \lambda_{\alpha i}^{(c)} c_i, \sum_{j\,\text{used by}\,\alpha} \lambda_{\alpha j}^{(n)} n_j \right)$$

Here, the sum over *i* (respectively *j*) is carried out over all carbon (nitrogen, respectively) sources that this species is capable of converting to its biomass. One may also consider the possibility of diauxic shifts between substitutable nutrient sources. In this case, each generalist species is following a predetermined preference list of nutrients and uses its carbon and nitrogen resources one-at-a-time, as modelled in *Goyal et al. (2018)*. Since at any state each of the species is using a 'specialist strategy', that is to say, it is growing on a single carbon and a single nitrogen source, we expect that many of the results of the current study would be extendable to this model. Interestingly, the stable marriage problem can be used to predict the stable states of microbial communities with diauxic shifts between substitutable resources (*Goyal et al., 2018*) and those in communities growing on a mix of two essential nutrients as in this study. It must be pointed out that these models use rather different variants of the stable marriage model.

It is straightforward to generalize our model to Monod's growth equation and to take into account non-zero death rate (or maintenance cost) of individual species (see Appendix 1).

One can extend our model to include cross-feeding between the species. In this case some of the nutrients are generated as metabolic byproducts by the species in the community. These byproducts should be counted among nutrient sources and thus would allow the number of species to exceed the number of externally supplied resources.

Above we assumed a fixed size of the species pool. This constraint could be modified in favor of an expanding pool composed of a constantly growing number of species. These new species correspond to either migrants from outside of the community or mutants of the species within the community. This variant of the model would allow one to explore the interplay between ecosystem's maturity (quantified by the number of species in the pool) and its properties such as multistability and propensity to regime shifts.

## Control of microbial ecosystems exhibiting multistability and regime shifts

In many practical situations we would like to be able to control microbial communities in a predictable and robust manner. That is to say, we would like to be able to reliably steer the community into one of its stable states and to maintain it there for as long as necessary. Alternative stable states and regimes shifts greatly complicate the task of manipulation and control of microbial ecosystems. Indeed, multistability means that the environmental parameters alone do not fully define the state of the community. In order to get it to a desired state, one needs to carefully select the trajectory along which one changes the environmental parameters (nutrient supply rates). Changing these parameters could lead to disappearance (local extinction) of some microbial species and open the ecosystem for colonization by others thereby changing its state. However, not all the states could be directly converted to each other in one step due to them being restricted to vastly different environments. Thus, densely interconnected networks of regime shifts shown in *Figure 2E–F* can be viewed as maps guiding the selection of the optimal environmental trajectory leading to the desired stable species composition.

These maps also suggest that microbial ecosystems described by our model might have a relatively small number of key drivers of regime shifts roughly corresponding to network modules (see *Figure 2F*). Regime shifts in each of the modules are driven by the competition between two mutually exclusive sets of keystone species. In addition to these keystone species, states also include 'satellite' species that do not generally affect the bistable switch. The exploration of different manipulation strategies of microbial ecosystems and the role of keystone and peripheral species in regime shifts is the subject of our future research (Maslov et al., unpublished).

## Materials and methods

### Identification of all states and classification of them as invadable or uninvadable

The competitive exclusion principle states that, in general, two species competing for the same growth-limiting nutrient cannot coexist with each other. Accounting for non-limiting nutrients present in our model, the competitive exclusion principle can be reformulated as the following two rules:

- Rule 1: In a given steady state each nutrient (either carbon or nitrogen) limits the growth of no more than one species.
- Rule 2: Any number of species can use a given nutrient in a non growth-limiting fashion. However, each of such species needs to be able to survive at the steady state concentration of this nutrient set by the growth-limited species. That means that for every nutrient each of the non growth-limited species $\beta$ needs to be more competitive than the grow-limited species $\alpha$ for the same resource: $\lambda_\alpha^{(c)} < \lambda_\beta^{(c)}$ (or $\lambda_\alpha^{(n)} < \lambda_\beta^{(n)}$ in case of a nitrogen nutrient).

Note that in any state of our model every species has a unique nutrient limiting its growth. By the virtue of the Rule 1, if a nutrient is limiting the growth of any species at all, such species is also unique. Hence, in a given state the relationship between surviving species and their growth-limiting nutrients (marked as shaded ovals in *Figure 1A*) is an example of a matching on a graph of resource utilization. Rule two imposes additional limitations on this matching. As we show in the Appendix 2 , uninvadable states correspond to stable matchings in a variant of the celebrated stable marriage problem (*Gale and Shapley, 1962*; *Gusfield and Irving, 1989*).

Just like in the MacArthur model (*MacArthur and Levins, 1964*) or any other consumer-resource model for that matter, the number of species present in a steady state of the community cannot exceed the total number of nutrients they consume. Any community constructed using Rules 1 and 2 represents a steady state of the ecosystem feasible for a certain range of nutrient supply rates. This state can be either invadable or uninvadable, and either dynamically stable or not.

For simplicity in our simulations we use equal numbers of C and N resources ($L$ carbons and $L$ nitrogens), with one unique species capable of utilization of every pair of resources ($L^2$ species in total). One must reiterate that our theory is not restricted to the specific values of $K$, $M$, and $S$. We first selected the values of $\lambda_{(i,j)}^{(c)}$ and $\lambda_{(i,j)}^{(n)}$ from a uniform random distribution between 10 and 100. Note that in the regime of high nutrient supply ($\phi^{(c,n)} >> \frac{\delta^2}{\lambda^{(c,n)}}$) all steady states of the community can be identified and tested for invadability using only the relative rank order of species' competitiveness for nutrients. For this we used the following exhaustive search algorithm:

*Step 1* - Select the subset of species whose growth is limited by C (C-limited species). For every carbon nutrient there are $L$ ways to choose a species using this nutrient. There is also an additional possibility that this nutrient is not limiting the growth of any species. Thus, the total number of possibilities is $L + 1$ for each of $L$ carbon nutrients. There are $(L+1)^L$ ways to choose the set of C-limited species and our algorithm will exhaustively investigate each of these potential steady states one-by-one.

*Step 2* - Given the set of C-limited species selected in Step 1, we now select the set of N-limited species. We first eliminate from our search any species that doesn't have enough carbon to grow. That is to say, we go over all carbon nutrients one-by-one and eliminate all species whose $\lambda^{(c)}$ is smaller than that of the C-limited species (if any) for this carbon nutrient. Among the remaining species we go over the nitrogen nutrients one-by-one and look for all possible ways to add a species limited by a given nitrogen source $n_j$ that satisfy the Rule 2. More specifically, we identify all species that use $n_j$ and can grow on their carbon sources (those are the only species that remained after the elimination procedure described above). We then compare $\lambda^{(n)}$s of these species to $\lambda^{(n)}$s of all C-limited species using $n_j$. To satisfy the Rule 2 for each $n_j$ we can add at most one N-limited species and its $\lambda^{(n)}$ has to be smaller than $\lambda^{(n)}$s of all C-limited species using $n_j$. Let $M_j$ be the number of such species ($M_j = 0$ if there are no such species for a given $n_j$). The total number of possible steady states of our model for a given combination of C-limited species selected in Step 1 is given by $\prod_{j=1}^{L}(M_j + 1)$. Here the $+1$ factor in $M_j + 1$ takes into account an additional possibility to not add any N-limited species for $n_j$.

The unique way to construct an uninvadable state by following this algorithm is to go over all nitrogen sources one-by-one and for each of them add the N-limited species with the largest $\lambda^{(n)}$ among all species using this resource, whose growth is allowed by carbon constraints. If for every $n_j$ this species is allowed by the Rule 2, that is to say, if its $\lambda^{(n)}$ is smaller than $\lambda^{(n)}$ of all C-limited species using $n_j$, we successfully constructed a unique uninvadable state for a given set of C-limited species. Indeed, all possible invading species that are allowed to grow by their carbon nutrients will be blocked by their nitrogen nutrients. If, however, for any of $n_j$, the species with the largest $\lambda^{(n)}$ is not

allowed by the Rule 2, that is to say, if its $\lambda^{(n)}$ is larger than $\lambda^{(n)}$ of at least one of the C-limited species, this species would make a successful invader of any state we construct. In this case, there is no uninvadable state for the set of C-limited species selected during the Step 1.

We used the above procedure to identify all possible steady states and to classify them as invadable and uninvadable for different numbers of resources used in our 2C × 2N × 4S and 6C × 6N × 36S examples. Note that, while this method is computationally feasible for a relatively small number of nutrients (we were able to successfully use it for up to 9 nutrients of each type), for larger systems one should rely on computationally more efficient algorithms based on the stable marriage problem (*Gale and Shapley, 1962*; *Gusfield and Irving, 1989*) as described in the Appendix 3.

## Monte-Carlo sampling of nutrient supply rates to identify feasible ranges of states

Given the parameters defining all species (i.e., the set of their $\lambda$s and $Y$s) and the chemostat dilution constant $\delta$, each state $p$ is feasible within a finite region in the nutrient supply space (a $K + M$ dimensional space $\vec{\phi} = \{\phi_i^{(c)}, \phi_j^{(n)}\}$), where all microbial populations and nutrient concentrations are non-negative and the limiting nutrients of every surviving species do not change. It is easy to show that in a steady state our system satisfies mass conservation laws for each of the nutrients:

$$
\begin{aligned}
c_i &+ \sum_{\text{all } \alpha \text{ using } c_i} \frac{B_\alpha}{Y_\alpha^{(c)}} = \frac{\phi_i^{(c)}}{\delta} \quad, \\
n_j &+ \sum_{\text{all } \alpha \text{ using } n_j} \frac{B_\alpha}{Y_\alpha^{(n)}} = \frac{\phi_i^{(n)}}{\delta} \quad.
\end{aligned}
\tag{4}
$$

To simplify the process of calculating the feasible volumes of all states we worked in the *limit of high nutrient supply* where $\phi_i^{(c)} \gg \frac{\delta^2}{\lambda_\alpha^{(c)}}$ and $\phi_j^{(n)} \gg \frac{\delta^2}{\lambda_\alpha^{(n)}}$ for all species $\alpha$. In this case the concentration $\delta/\lambda_\alpha^{(c,n)}$ of any nutrient limiting growth of some species ($\alpha$ in this case) is negligible compared to its 'abiotic concentration' $\phi_i^{(c,n)}/\delta$, that is to say, its concentration before any microbial species were added to the chemostat. In this case one can ignore the terms $c_i$ and $n_j$ in *Equation 4* for all nutrient limiting growth of some species and leave only the ones that are not limiting the growth of any species. It is convenient to introduce the K + M − dimensional vector $\vec{X}_p$ of microbial abundances and non-limiting nutrient concentrations in a given state $p$. For example, for the uninvadable state #5 in the 2C × 2N × 4S model we have: $\vec{X}_5 = \{B_{(1,1)}, B_{(1,2)}, B_{(2,2)}, n_2\}$.

The mass conservation laws (*Equation 4*) can be used to obtain the feasible volumes of all states and can be represented in a compact matrix form for each state $p$:

$$
\vec{\phi} = \hat{R}_p \vec{X}_p \quad,
\tag{5}
$$

where $\phi$ is the vector of $K + M$ nutrient supply rates and $\hat{R}_p$ is a matrix composed of inverse yields $Y^{-1}$ of surviving species and '1' for each of the non-limiting nutrients in a given state $p$. For example, for the state #5 in our 2C × 2N × 4S model the *Equation 5* expands to:

$$
\begin{bmatrix} \phi_1^{(c)} \\ \phi_2^{(c)} \\ \phi_3^{(c)} \\ \phi_4^{(c)} \end{bmatrix} = \begin{bmatrix} \frac{1}{Y_{(1,1)}^{(c)}} & \frac{1}{Y_{(1,2)}^{(c)}} & 0 & 0 \\ 0 & 0 & \frac{1}{Y_{(2,2)}^{(c)}} & 0 \\ \frac{1}{Y_{(1,1)}^{(n)}} & 0 & 0 & 0 \\ 0 & \frac{1}{Y_{(1,2)}^{(n)}} & \frac{1}{Y_{(2,2)}^{(n)}} & 1 \end{bmatrix} \begin{bmatrix} B(1,1) \\ B(1,2) \\ B(2,2) \\ n_2 \end{bmatrix}.
\tag{6}
$$

Using *Equation 5,* it is easy to check if a given state is feasible at a particular nutrient supply rate $\vec{\phi}$ by multiplying $\hat{R}_p^{-1}$ (the inverse of the matrix $\hat{R}_p$) with $\vec{\phi}$. If all of the elements of the resulting vector $\vec{X}_p$ are positive, then the state $p$ is feasible at $\vec{\phi}$. If the matrix $\hat{R}_p$ is not invertible that is $\det(\hat{R}_p) = 0$, the state is feasible only on a low-dimensional subset of nutrient supply rates. This is not possible for a general choice of yields $Y$ and is not considered in our study.

The parameters $\lambda$ and $Y$ for 2C $\times$ 2N $\times$ 4S and 6C $\times$ 6N $\times$ 36S realizations of the model were drawn from the uniform random distributions and are listed in *Supplementary files 1,2* and *3,4,5,6*. For $\lambda$'s the distribution ranges between 10 and 100. For $Y$'s it is between 0.1 and 1.

In our numerical simulations, for each model realization, we sampled $10^6$ random nutrient supply rate combinations $\vec{\phi}$. Supply rates of each individual nutrient were independently selected from a uniform distribution on the [10, 1000] interval. We refer to this procedure as Monte Carlo sampling. The lower bound ensures that the system is always in the limit of high nutrient supply since $\max\left(\frac{\delta^2}{\lambda_\alpha}\right) = 0.1 \ll 10$..

Then we checked the feasibility of each of the 33 possible steady states (both invadable and uninvadable) in the 2C $\times$ 2N $\times$ 4S model and each of the 1211 uninvadable steady states in the 6C $\times$ 6N $\times$ 36S model. That is to say, for every set of nutrient supply rates $\vec{\phi}$ and for every state $p$ we checked whether all elements of $\vec{X}_p$ are positive. The feasible range of nutrient supply rates of each state was estimated as the fraction of nutrient supply rate combinations (out of 1 million vectors $\vec{\phi}$ sampled by our Monte Carlo algorithm) where this state was found to be feasible.

## The network of regime shifts from overlaps of feasible ranges

Two stable states are said to be capable of a regime shift if their feasibility ranges overlap with each other, that is if there exists at least one nutrient supply rate combination at which both these states are feasible. We used the data obtained by the Monte-Carlo sampling to look for such cases and to construct networks shown in *Figure 2E*, *Figure 2F*.

We performed additional simulations to look for boundaries between uninvadable states in our realization of the 2C $\times$ 2N $\times$ 4S model. In order to do that, for each state we generated a large ensemble of random vectors of bacterial abundances of surviving species and concentrations of all not-limiting nutrients. The population of one of the species (also randomly selected) was set to be a small negative number ($-0.01$). This represents continuous gradual extinction of this species upon crossing of the boundary, The populations of state's other surviving species and the concentrations of its non-limiting nutrients were drawn from the uniform distribution between (0,1]. Using *Equation 5,* we calculated the nutrient supply rates $\phi$ corresponding to this case. For these nutrient supply rates lying just across the feasibility boundary of the originally selected state, we checked the feasibility of other five uninvadable dynamically stable states. If any of these states ended up being feasible, we assumed that this state shares a boundary with the originally selected one. Using these procedure we found four bordering pairs of states shown as red edges in *Figure 2E*.

We used Gephi 0.9.2 software package to visualize the network in *Figure 2F* and to perform its modularity analysis. Seven densely interconnected clusters shown with different colors in *Figure 2F* were identified using Gephi's built-in module-detection algorithm (*Blondel et al., 2008*) with the resolution parameter set to 1.5.

## Dynamic stability of states

We checked the dynamic stability of every 33 possible steady state (both invadable and uninvadable) (for the 2C $\times$ 2N $\times$ 4S model) and each of the 1211 uninvadable states (for the 6C $\times$ 6N $\times$ 36S model) using the following two algorithms:

1. Small perturbation analysis
   For 2C $\times$ 2N $\times$ 4S example and each of the 33 states we selected many supply rate combination where this state is feasible. For each of these supply rates we generated the populations in our ecosystem to be equal to their steady state values. We then subjected them to small perturbations of of all nutrient concentrations and of all microbial populations of species present in the state. If after some transient period all populations and concentrations returned to their steady state values - the state was declared to be dynamically stable. If they drifted to these in some other steady state - the original state was declared to be dynamically unstable. Based on our numerical simulations the dynamical stability of the state was independent of the nutrient supply rates at which this numerical experiment has been performed. We choose to perturb only the populations of the species present in the state because an invadable state, by definition, would always be dynamically unstable against an addition of a very small population of at least one invading species from the species pool. This instability should not render it dynamically unstable. The numerical integration of the system dynamics following a

perturbation was done in C programming language using the CVODE solver library of the SUNDIALS package (*Hindmarsh et al., 2005*).

2. Inference of state's dynamic stability from the pattern of its overlaps with other states

The number of uninvadable states (1211) in our 6C × 6N × 36S model was too large to be directly tested for dynamic stability as we did for the 2C × 2N × 4S model. Their dynamic stability was instead inferred from our Monte-Carlo simulations listing all feasible uninvadable states for every sampled nutrient supply rate combination. We first identified 1022 uninvadable states, which were the unique feasible state for at least one nutrient supply point. All such states should be dynamically stable, since for every nutrient supply rate there should be at least one uninvadable dynamically stable state representing the end point of system's dynamics. The remaining 173 uninvadable states, which were feasible for at least one of 1 million sampled nutrient supply rates were labelled as potentially dynamically unstable. Note that in our Monte-Carlo analysis we only sampled a finite (albeit large) number of nutrient supply rate combinations. Thus it is entirely possible that we missed some crucial supply rate combinations for which one of these states was the only uninvadable state. Any such point would have rendered this state as dynamically stable. Such false assignments might lead to a violation of the basic empirical rule in our model stating that $V$ uninvadable stable states are always accompanied by $V − 1$ uninvadable dynamically unstable states ($V/(V − 1)$ rule) for some sampled nutrient supply rates. In our Monte Carlo simulations of the 6C × 6N × 36S model the $V/(V − 1)$ rule was violated for only 370 nutrient supply rates combinations out of 1,000,000 sampled points. We believe that these violations were caused by an incorrect identification of dynamically unstable states mentioned above. To iteratively refine the lists of stable and unstable states, we went over all potentially unstable states one-by-one and checked whether reclassifying the state involved in the largest number of violations as stable would reduce the overall number of violations. If it did, we reclassified this state as stable and recalculated the number of violations for all remaining points. By the end of this iterative procedure we were able to completely eliminate violations by reassigning 36 potentially unstable states as dynamically stable. This left us with 1022 + 36 = 1058 dynamically stable and 173 − 36 = 137 dynamically unstable uninvadable states in the 6C × 6N × 36S model. The remaining 1211 − 1058 − 137 = 16 uninvadable states were not feasible for any of 1,000,000 sampled nutrient supply rates. Hence their dynamic stability remains unidentified. Both 36 reassigned states and 16 undetected states are expected to have very small ranges of feasible nutrient supply rates.

## Multistability as a function of variation in stoichiometric ratios of different species

To investigate how multistability in our model depends on variation in stoichiometric ratios of different species, we simulated 4000 variants of the 2C × 2N × 4S model. In these variants, we kept the same choice of species competitiveness (quantified by their $\lambda$'s) but reassigned their yields $Y$. To cover a broad range of standard deviations of N:C stoichiometry of different species (their $Y_\alpha^{(c)}/Y_\alpha^{(n)}$) we randomly sampled yield combinations from gradually expanding intervals. First we simulated 1000 model variants, where yields of four species were independently drawn from uniform distribution $U(0.45, 0.55)$. These simulations were followed by 1000 model variants, where yields of four species were drawn from $U(0.3, 0.7)$, 1000 model variants with yields from $U(0.1, 0.9)$ and, finally, 1000 model variants with yields from $U(0.01, 1.0)$. In each variant of the model with a particular set of yields of four species, we calculated the fraction of multistable points among $10^5$ nutrient supply rate combinations as described in the section 5.2 Monte-Carlo sampling of nutrient supply rates to identify feasible ranges of states of Materials and methods. The results are shown in *Figure 5*.

## GitHub repository of the code used in our project

The PCA analysis, plots and statistical tests were implemented using R version 3.4.4. Other simulations were carried out in C (using compiler gcc version 5.4.0) and Python 3.5.2. Matlab analysis was done using MATLAB and Statistics Toolbox Release 2018a, The MathWorks, Inc, Natick, Massachusetts, United States. The code for both our simulations and statistical analysis can be downloaded from: https://github.com/ssm57/CandN (*Dubinkina and Maslov, 2019*; copy archived at https://github.com/elifesciences-publications/CandN).

## Acknowledgements

Part of this work has been carried out at the University of Padova, Italy, in August 2018, during a scientific visit by one of us (SM). The authors wish to thank Prof. James O'Dwyer, Prof. Seppe Kuehn and Akshit Goyal for a critical reading of an earlier version of the manuscript. The authors also thank reviewers and editors for their comments, that helped to improve the manuscript.

## Additional information

### Funding
No external funding was received for this work.

### Author contributions
Veronika Dubinkina, Software, Investigation, Visualization, Writing—original draft, Writing—review and editing; Yulia Fridman, Conceptualization, Methodology, Writing—original draft, Writing—review and editing; Parth Pratim Pandey, Software, Investigation, Methodology, Writing—review and editing; Sergei Maslov, Conceptualization, Software, Supervision, Funding acquisition, Investigation, Visualization, Methodology, Writing—original draft, Project administration, Writing—review and editing

### Author ORCIDs
Veronika Dubinkina https://orcid.org/0000-0002-4844-6795
Sergei Maslov https://orcid.org/0000-0002-3701-492X

### Decision letter and Author response
Decision letter https://doi.org/10.7554/eLife.49720.SA1
Author response https://doi.org/10.7554/eLife.49720.SA2

## Additional files

### Supplementary files
• Supplementary file 1. Values of species' competitive abilities for carbon and nitrogen in $2C \times 2N \times 4S$ model.

• Supplementary file 2. Values of species' carbon and nitrogen yields in $2C \times 2N \times 4S$ model.

• Supplementary file 3. Values of species' competitive abilities for carbon in $6C \times 6N \times 36S$ model.

• Supplementary file 4. Values of species' competitive abilities for nitrogen in $6C \times 6N \times 36S$ model.

• Supplementary file 5. Values of species' carbon yields in $6C \times 6N \times 36S$ model.

• Supplementary file 6. Values of species' nitrogen yields in $6C \times 6N \times 36S$ model.

• Transparent reporting form

### Data availability
All data generated or analysed during this study are included in the manuscript and supporting files. The code for both our simulations and statistical analysis can be downloaded from: https://github.com/ssm57/CandN (copy archived at https://github.com/elifesciences-publications/CandN).

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

## Appendix 1

## General form of growth laws

It is straightforward to generalize our model to allow a more general functional form for growth laws than Liebig's law, $\min(\lambda_\alpha^{(c)} c_i, \lambda_\alpha^{(n)} n_j)$. Microbial growth on two essential substrates is thought to normally follow Monod's equation for the rate-limiting nutrient: $g_\alpha^{(m)} \min(c_i/(K_\alpha^{(c)} + c_i), n_j/(K_\alpha^{(n)} + n_j))$ (See **Kovárová-Kovar and Egli, 1998** for a discussion of limitations of Monod's law). For low concentrations of the rate-limiting nutrient, say carbon source, the Monod's law simplifies to the linear growth law used throughout this study: $g_\alpha = \lambda_\alpha^{(c)} c_i$. Microbes' competitive abilities, also known as their specific affinities towards each substrate, are related to the parameters of Monod's law via

$$\lambda_\alpha^{(c)} = \frac{g_\alpha^{(m)}}{K_\alpha^{(c)}}; \quad \lambda_\alpha^{(n)} = \frac{g_\alpha^{(m)}}{K_\alpha^{(n)}} \tag{S1}$$

In another variant of growth laws, two essential nutrients at low concentrations jointly affect the growth rate of the microbe: $g_\alpha^{(m)} c_i \cdot n_j / [(K_\alpha^{(c)} + c_i) \cdot (K_\alpha^{(n)} + n_j)]$ (see **Bader, 1978** for a discussion of these and other forms of double-substrate growth law). For simplicity of mathematical calculation we limited this study to Liebig's law. However, many of the essential results we obtained (e. g. possible multistability in a system where species have different yields) hold for any growth laws listed above. In fact, the low concentration version of the previous growth law, where $g_\alpha^{(m)} c_i \cdot n_j$ has been studied by one of us in the context of autocatalytic growth of heteropolymers (**Tkachenko and Maslov, 2018**). Instead of exponentially replicating microbial species **Tkachenko and Maslov (2018)** considers pairs of mutually catalytic (and thus exponentially growing) complementary '2-mers' (a specific sequence of two consecutive monomers anywhere within a polymer chain). This minor difference complicates the math, while leaving the basic properties unchanged. Just like in our system, where up to $2L$ species (out of $L^2$ candidates) may simultaneously survive in the steady state of an ecosystem grown on of $L$ carbon and $L$ nitrogen sources, the polymer systems have no more than $2Z$ 2-mer 'species' (out of $Z^2$ candidates) surviving in the steady state with polymers having $Z$ possible monomers on their right ends and $Z$ possible monomers on their left ends. Many (but not all) results of this paper are largely consistent with the present study. Note that for polymers the yields of all 'species' are equal to 1, that is to say, one new 2-mer is formed upon ligation of one left end of a polymer with one right of another polymer chain. Yet, the model in **Tkachenko and Maslov (2018)** is capable of (at least) bistability. At present, it is not clear if this is due to autocatalytic cycles having length two or this property would survive in a simpler version of the model in which instead of the Equation (1) of **Tkachenko and Maslov (2018)** one has

$$\dot{d}_{ij} = d_{ij}(\lambda_{ij} l_i r_j - \delta)$$

and the overall fluxes of left and right ends are independent from each other (instead of both being equal to $c_i = \phi_i/\delta$ as in **Tkachenko and Maslov, 2018**).

Another variant of the model is where each species $\alpha$ has its own unique 'death' or 'maintenance' rate $\delta_\alpha$, playing the role of the same dilution rate $\delta$. The steady states of this model (but not the dynamics leading to these states) can be calculated by dividing both sides of **Equations 2** by $\delta_\alpha$. This is equivalent by redefining the competitiveness parameters to $\tilde{\lambda}_\alpha = \lambda_\alpha/\delta_\alpha$ and setting the chemostat dilution rate to $\tilde{\delta} = 1$. All of our results in the high-flux regime $\phi \gg \delta^2/\lambda$ would remain unchanged.

From (**Equation 5**-**Equation 6**) one can see that when all species have the same C:N stoichiometry, the maximal number of microbialspecies in a state is equal to the number of nutrients minus 1. Indeed, one can show that a state $p$ with $S_{surv} = K + M$ has $\det(\hat{R}_p) = 0$, which means that the feasible volume of any such state is zero. These states are only possible

on a lower-dimensional manifold in the $(K + M)$-dimensional space of supply rates (these results have been already discussed by Tilman in his special case; *Tilman, 1982*).

Multistability is also possible in a variant of the MacArthur model (*MacArthur and Levins, 1964*; *MacArthur, 1970*; *Chesson, 1990*) in which different species have different yields on individual carbon sources (Maslov, unpublished). A convex Lyapunov function (*MacArthur, 1970*) precluding multistability does not exist in this case.

## Appendix 2

# Constraints on steady states from microbial and nutrient dynamics

A steady state of equations describing the microbial dynamics (*Equation 2*) is realized when either $B_\alpha = 0$ (the species was absent from the system from the start or subsequently went extinct) or when its growth rate $g_\alpha$ is exactly equal to the chemostat dilution rate $\delta$. This imposes constraints on steady state nutrient concentrations with the number of constraints equal to the number of microbial species present with non-zero concentrations. Since, in general, the number of constraints cannot be larger than the number of constrained variables, no more than $K + M$ of species could be simultaneously present in a steady state of the ecosystem. For Liebig's growth law used in this study, each resource can have no more than one species for which this resource limits its growth, that is to say, which sets the value of the minimum in $\min(\lambda_\alpha^{(c)} c_i, \lambda_\alpha^{(n)} n_j)$ The steady state concentrations of these resources are given by $c_i^{(*)} = \delta / \lambda_\alpha^{(c)}$ (if the growth is limited by the carbon source) and $n_j^{(*)} = \delta / \lambda_\alpha^{(n)}$ (if the growth is limited by the nitrogen source). Here $\alpha$ is the species whose growth is rate-limited by the resource in question. In a general case, no more than one species can be limited by the same resource (carbon in our example), since the species with the largest $\lambda^{(c)}$ would outcompete other species with smaller values of $\lambda^{(c)}$ by making the steady state concentration $c_i^{(*)}$ so low that other species can no longer grow on it. Note however, that multiple species $\beta$ could consume the same resource as the rate-limiting species $\alpha$, as long as their growth is not limited by the resource. Each of these species must then be limited by their other nutrient (a nitrogen source in our example). However, their survival requires that carbon concentration set by species $\alpha$ is sufficient for their growth. Thereby, any species growing on a resource in a non-limited fashion must have $\lambda_\beta^{(c)} > \lambda_\alpha^{(c)}$.

Mathematically, it cane be proven by observing that, since species $\beta$ is limited by its nitrogen resource, one must have $\lambda_\beta^{(c)} c_i^{(*)} > \lambda_\beta^{(n)} n_j^{(*)}$. At the same time in a steady state, the concentrations of all rate-limiting resources are determined by the dilution rate $\delta$ via $\lambda_\beta^{(n)} n_j^{(*)} = \delta$, and $\lambda_\alpha^{(c)} c_i^{(*)} = \delta$. Combining the above three expressions one gets: $\lambda_\beta^{(c)} c_i^{(*)} > \lambda_\beta^{(n)} n_j^{(*)} = \delta = \lambda_\alpha^{(c)} c_i^{(*)}$, or simply $\lambda_\beta^{(c)} > \lambda_\alpha^{(c)}$. The constraints on competitive abilities $\lambda$ for species present in a steady state in our model are then:

- Exclusion Rule 1: Each nutrient (either carbon or nitrogen source) can limit the growth of no more than one species $\alpha$. From this it follows that the number of species co-existing in any given steady state cannot be larger than $K + M$, the total number of nutrients.
- Exclusion Rule 2: Each nutrient (e.g. specific carbon source) can be used by any number of species in a non-rate-limiting fashion (that is to say, where it does not constrain species growth in Liebig's law). However, any such species $\beta$ has to have $\lambda_\beta^{(c)} > \lambda_\alpha^{(c)}$, where $\lambda_\alpha^{(c)}$ is the competitive ability of the species whose growth is limited by this nutrient. In case of a nitrogen nutrient, the constraint becomes $\lambda_\beta^{(n)} > \lambda_\alpha^{(n)}$.

Note that the steady state solutions of equations *Equation 2* do not depend on populations $B_\alpha$ of surviving species. Their steady state populations $B_\alpha^{(*)}$ are instead determined by *Equation 3*. Taking into account that, in a steady state, the growth rate of each surviving species is exactly equal to the dilution rate $\delta$ of the chemostat, after simplifications one gets:

$$\frac{\phi_j^{(c)}}{\delta} = c_i^{(*)} + \sum_{\text{all } \alpha \text{ using } c_i} \frac{B_\alpha^{(*)}}{Y_\alpha^{(c)}}$$

$$\frac{\phi_j^{(n)}}{\delta} = n_j^{(*)} + \sum_{\text{all } \alpha \text{ using}} n_j \frac{B_\alpha^{(*)}}{Y_\alpha^{(n)}}$$

(S2)

As described above, the steady state concentration of resources are given by $\delta/\lambda_\alpha^{(\text{corn})}$, where $\alpha$ are the species rate-limited by each resource. In the absence of such species, the concentration of a resource is given by anything left after it being consumed by surviving species in a non-rate-limiting manner. One can show that in this case, the resource (e.g. carbon) concentration has to be larger than $\delta/\lambda_\beta^{(c)}$, where $\lambda_\beta^{(c)}$ is the smallest affinity among microbes utilizing this resource.

One convenient approximation greatly simplifying working with *Equation S2* is the 'high-flux limit' in which $\phi_i^{(c)} \gg \delta^2/\lambda_\alpha^{(c)}$ and $\phi_j^{(n)} \gg \delta^2/\lambda_\alpha^{(n)}$. In this approximation one can approximately set to zero the steady state concentrations of all resources that have a species rate-limited by them. The steady state concentrations of the remaining resources can take any value as long as it is positive. Hence, in this limit the *Equation S2* can be viewed as a simple matrix test of whether a given set of surviving species limited by a given set of resources is possible for a given set of nutrient fluxes. Indeed, my multiplying the vector of fluxes with the inverse of the matrix $\hat{R}$ composed of inverse yields of surviving species and one for nutrients not limiting the growth of any species one formally gets the only possible set of steady state species abundances, $B_\alpha^{(*)}$, and a subset of non-limiting resource concentrations $c_i^{(*)}$ and $n_j^{(*)}$. If all of them are strictly positive - the steady state is possible. If just one of them enters the negative territory - the steady state cannot be realized for these fluxes of nutrients.

The above rule can be modified to apply even below the high-flux limit with the following modifications: 1) Instead of $\phi^{(c)}$ (or $\phi^{(n)}$), one uses their 'effective values' $\tilde{\phi}^{(c)}$ (or $\tilde{\phi}^{(n)}$) introduced in *Goyal et al. (2018)*, determined as

$$\tilde{\phi}_i^{(c)} = \phi_i^{(c)} - \frac{\delta^2}{\lambda_{\alpha(i)}^{(n)}}$$

$$\tilde{\phi}_i^{(n)} = \phi_i^{(n)} - \frac{\delta^2}{\lambda_{\alpha(i)}^{(c)}} \quad ,$$

(S3)

where $\alpha(i)$ is the (unique) species limited by the nutrient $i$. If the nutrient is not limiting for any os the species in the steady state, $\alpha(i)$ is the species using the nutrient in a non-limited fashion, which has the *smallest* value of $\lambda$. This last rule comes from the observation that in order for a non-limiting resource not to become limiting for a species $\beta$ currently using it in a non-limiting fashion, its concentration cannot fall below $\delta/\lambda_\beta^{(x)}$. Thus, when checking the feasibility of a given state, the concentration of a non-limiting resource can be written as $\delta/\lambda_\beta^{(x)} +$ a positive number, or (more conveniently) the influx of this resource can be offset as described in *Equation S3*

## Appendix 3

### Stable matching approach for identification and classification of steady states

First we describe the exact one-to-one mapping between all uninvadable steady states (UIS) in our model and the complete set of 'stable marriages' in a variant of a well-known stable marriage or stable allocation problem developed by Gale and Shapley in the 1960s (*Gale and Shapley, 1962*) and awarded the Nobel prize in economics in 2012. This mapping provides us with constructive algorithms to identify and count all uninvadable steady states in our ecosystem.

We start by considering a special case of our problem with $L$ carbon and $L$ nitrogen sources and a pool of $L^2$ species, such that for every pair of sources $c_i$ (carbon) and $n_j$ (nitrogen) there is exactly one microbe $B_{ij}$ capable of using them. For the sake of simplicity we have switched the notation from $B_\alpha$ to $B_{ij}$, where $\alpha = (ij)$ is the unique microbe in our pool capable of growing on $c_i$ and $n_j$. Having considered this simpler situation we will return to the most general case of unequal numbers of carbon ($K$) and nitrogen ($M$) resources and any number of microbes from a pool of $S$ species competing for a given pair of resources.

This is where we need to revise in certain ways the network representation of a steady state used in the main text (see *Figure 1A*). In the marriage game related theory, the notion of (stable or unstable) matching explicitly refers to a bipartite graph with two distinct sets of vertices and edges arranged in such a way that each one may join only a pair of elements belonging to different sets. In our case, it is natural to consider two sets of resource nodes (vertices), one including all carbon nodes and the other one containing all nitrogen nodes. An edge, or link, will appear between a carbon $c_i$ and nitrogen $n_j$ node if the microbe $B_{ij}$ using these two nutrients is present in the state represented by this particular bipartite network.

Furthermore, the specifics of our version of 'marriage game', or rather 'residents vs hospitals', problem requires us to consider *directed* bipartite graphs as the steady state representations in our model. For any species $B_{ij}$ present in a given state, we choose the direction of the edge joining node $c_i$ with node $n_j$ to be pointing from $c_i$ to $n_j$ if the microbe is limited by its carbon nutrient (and the other way around, from $n_j$ to $c_i$, on the case of $B_{ij}$ being nitrogen-limited). *Appendix 3—figure 1A* shows the directed bipartite graph representation of the state #5 of a particular example of 2C $\times$ 2N $\times$ 4S system considered in the 'Results' section of the main text.

In what follows we will refer to a resource as *occupied* if in a given steady state there is a microbe for which this resource is rate-limiting. In our bipartite network representation occupied resources have an outgoing edge (their out-degree is equal to 1), while unoccupied resources have out-degree equal to 0.

### Review of results about stable matchings in the hospitals/residents problem

The hospitals/residents problem (*Gale and Shapley, 1962*) is known in various settings. The one directly relevant to our problem is the following. There are $L$ applicants for residency positions in $H \leq L$ hospitals. A hospital number $i$ has $V_i$ vacancies for residents to fill, $V_i$ ranging from zero to $L$, $\sum V_i = L$. Each hospital has a list of preferences in which residency applicants are strictly ordered by their ranks, from 1 (the most desirable) to $L$, (the least desirable). These lists are generally different for different hospitals. Each applicant has a ranked list of preferred hospitals ranging from 1 (the most desirable) to $H$ (the least desirable). Those lists can also vary between applicants. A *matching* is an assignment of applicants to hospitals such that all applicants got residency and all hospital vacancies are filled. A matching is *unstable* if there is at least one applicant $a$ and hospital $h$ to which $a$ is not assigned such that:

1.    Condition 1. Applicant $a$ prefers hospital $h$ to his/her assigned hospital;

2.    Condition 2. Hospital $h$ prefers applicant $a$ to at least one of its assigned applicants.

If such a pair $(a, h)$ exists, it is called 'a blocking pair' or 'a pair that blocks the matching'. A *stable* matching by definition has no blocking pairs. Gale and Shapley proved that for any set of applicant/hospital rankings and hospital vacancies there is at least one stable matching (**Gale and Shapley, 1962**). Generally the number of stable matching is larger than one. For example, for stable marriages and random rankings the average number of stable matchings is given by $L/e \log L$(**Gusfield and Irving, 1989**). To the best of our knowledge, the dependence of this number on the distribution of hospital vacancies has not been investigated. The fact that the actual number of uninvadable states is rather close to its lower bound indicates that, at least for $L \leq 9$, the number of stable matchings averaged over all possible in-degree allocations is rather close to 1.

Gale and Shapely not only proved the existence of at least one stable matching, but also proposed a constructive algorithm on how to find it. Listed below are the main steps in this algorithm optimized for for applicants. Each applicant first submits his/her application to the hospital ranking 1 in his/her preference lists. Each hospital considers all applications it received so far and accepts all of the applicants if their number is less or equal than hospital's announced number of vacancies, $L_i$. If the number of applicants exceeds $L_i$, the hospital gives a conditional admission to the best-ranking $L_i$ applicants according to hospital's own preference list. Each applicant not admitted to their top hospital goes a step down on his/her preference list and applies to the second-best hospital. The latter admits this applicant if (1) this hospital has not yet filled all of its vacancies or (2) all vacancies are filled, but among the conditionally admitted applicants there is at least one who ranks lower (according to hospital's list) than the new applicant. Such lower-ranked applicants are declined admission and replaced with better ones. They subsequently lower their expectations and apply to the next hospital on their list. After a number of iterations all applicants are admitted and all vacancies are filled so that this process stops. As **Gale and Shapley (1962)** proved, the resulting matching is stable. Furthermore, the theorem states that in this matching every applicant gets admitted to the best hospital among all stable matchings, while every hospital gets the worst set of residents among all stable matchings. Later research described in **Gusfield and Irving (1989)** describe more complex constructive algorithms allowing one to efficiently find all of the stable matchings starting with the applicant-optimal one.

Well developed mathematical apparatus of stable matching problem provides an invaluable help in the task of identifying all uninvadable states in microbial ecosystems. Indeed, without its assistance this task would require exponentially long time. To connect the problem of finding all uninvadable states to that of finding all stable matchings between hospitals and residents, we start with the following three observations:

1.    In any uninvadable steady state, either all carbon sources or all nitrogen sources (or both) are occupied. Indeed, if in a steady state a carbon source $c_i$ and a nitrogen source $n_j$ are not-limiting to any microbes, then microbe $B_{ij}$ can always grow and thereby invade this state. Thus uninvadable states can be counted separately: one first counts the states where all nitrogen sources are occupied, and then counts those in which all carbon sources are occupied. Double counting happens when both carbon and all nitrogen sources are occupied. We will keep the possibility of double counting in mind and return to this problem later.

2.    For a pool of species, where for every pair of resources there is exactly one microbe using each (carbon, nitrogen) pair, one can think of each of $L$ carbon (alternatively, nitrogen) sources as if it had a list of 'preferences' ranking all nitrogen (correspondingly carbon) sources. Indeed, the ranking of competitive abilities $\lambda_{ik}^{(c)}$ of different microbes using the same carbon source $c_i$ but different nitrogen sources $n_k$ can be viewed as the ranking of nitrogen sources $k$ by the carbon source $i$. Conversely, the ranking of $\lambda_{mj}^{(n)}$ with the same $n_j$ but variable $c_m$ can be thought of as ranking of carbon sources $c_m$ by the nitrogen source $n_j$.

3.    Consider a steady state in which all nitrogen sources are occupied. In our network representation it corresponds to every nitrogen source sending an outgoing link to some carbon source. Let $L_i$ be the number of microbes using the carbon source $i$ in a non-limiting

fashion (the in-degree of these outgoing links ending on $c_i$, see *Appendix 3—figure 1C*). Then, obviously, $L = \sum L_i$ (note that some of the terms in this sum might be equal to zero).

One can prove that if the state is uninvadable, then the matching given by all edges going from nitrogen sources to carbon sources must be stable in the Gale-Shapley sense. To prove this, let's think of nitrogen sources as 'applicants' and carbon sources as 'hospitals' with their numbers of 'vacancies' given by $L_i$. Indeed, any unstable matching has at least one blocking pair $(n_j, c_i)$ such that:

- Condition 1. The nitrogen source ('applicant') $n_j$'prefers' the carbon source ('hospital') $c_i$ to its currently assigned carbon source (the one used by the current microbe $B_{kj}$ limited $n_j$). This means that $\lambda_{ij}^{(n)} > \lambda_{kj}^{(n)}$. Thus the microbe $B_{ij}$ can grow on its nitrogen source (provided that it can also grow on its carbon source).
- Condition 2. The carbon source ('hospital') $c_i$'prefers' the nitrogen source ('applicant') $n_j$ to at least one of $L_i$ of its currently assigned carbon sources (the set of microbes using $c_i$ in a non-rate-limiting fashion). Thereby $\lambda_{ij}^{(c)}$ must be larger than the smallest $\lambda^{(c)}$ among these microbes. According to the Exclusion Rule 2, this smallest $\lambda^{(c)}$ is still larger than $\lambda^{(c)}$ of the microbe limited by $c_i$ (if it exists). Thus the microbe $B_{ij}$ can also grow on its carbon source.

This proves that the microbe $B_{ij}$ corresponding to any blocking pair can grow on both its carbon and its nitrogen sources, and thereby can successfully invade the steady state. This finishes the proof that any uninvadable state has to be a stable matching in the Gale-Shapley sense.

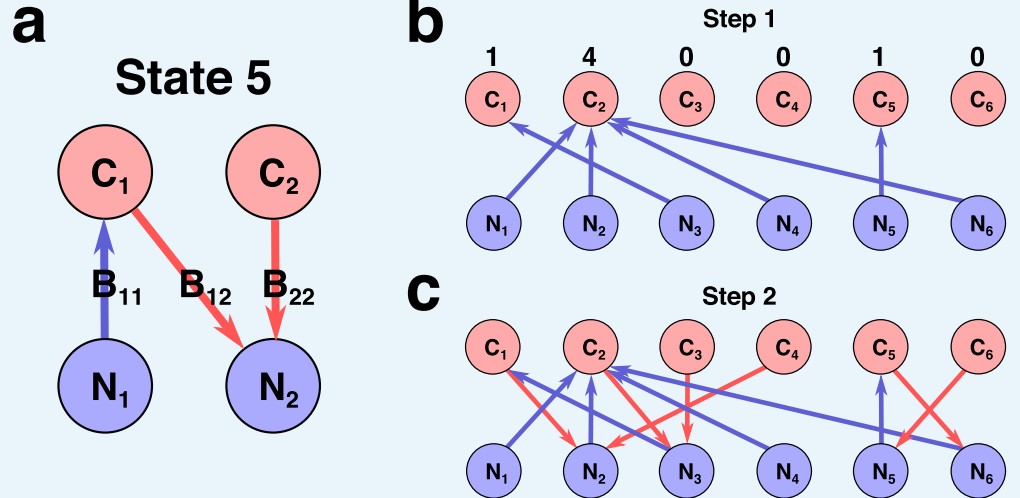

**Appendix 3—figure 1.** Network representation of a state in Stable Marriage analogy. (**a**) Schematic representation of state #5 in the 2C × 2N × 4S model (the same state is shown in *Figure 1A* in Main text). Here each species represented as an arrow connecting two resources it is utilizing for growth, with the color and direction of arrow representing growth-limitation of a species (red corresponds to C-limited species, blue N-limited one). (**b**)-(**c**) Schematic representation of two-step construction of state #991 in the 6C × 6N × 36S model. (**b**) We first assign all species that are growth-limited by N (blue links outgoing from N sources). The numbers above C sources indicate number of vacancies for a given resource. (**c**) Then for a given set of N-limited species we populate the remaining C-limited ones that are allowed by the Condition 2 (red links outgoing from C sources).

However, this does not prove that any stable matching corresponds to exactly one uninvadable state. To prove this we first notice that, up to this point, our candidate uninvadable state contained only the nitrogen-limited species (See *Appendix 3—figure 1B*) . We will now supplement it with carbon-limited species in such a way that (1) added species do

not violate the exclusion rule 2; (2) added species render the state completely uninvadable. Let is introduce a new notation (applicable to our case in which all nitrogen sources are occupied). Let $\lambda_{min}^{(c)}(i)$ denote the smallest $\lambda^{(c)}$ among all species using $c_i$ in a non-rate-limiting fashion. The Gale-Shapley theorem only guarantees the protection of our state from invasion by a species $(i,j)$ with $\lambda_{ij}^{(c)}$ larger than $\lambda_{min}^{(c)}(i)$ (see the Condition 2 above). To ensure that our state is uninvadable by the rest of the species, one needs to add some carbon-limited species to this state. In order to do this in a systematic way, for each $c_i$ we compile the list of all species using this carbon source with $\lambda^{(c)} < \lambda_{min}^{(c)}(i)$. Each of these species is a potential invader. Some species could be crossed off from the list of potential invaders because they cannot grow on their nitrogen source. These species have $\lambda^{(n)}$ below that of the (unique) species limited by their nitrogen source. Among the species that remained on the list of invaders after this procedure, we select that with the largest $\lambda^{(c)}$ and add it to our steady state as a $C \rightarrow N$ directed edge, that is to say, as a carbon-limited species. This will prevent all other potential invaders on our list, since they have smaller $\lambda_{(c)}$ and thus, following the addition of our top carbon-limited species, they would no longer be able to grow based on their carbon source. We will go over all $c_i$ and add such carbon-limited species if they are needed. The only scenario when such species is not needed if our list of potential invaders would turn up to be empty. In this case we will leave this carbon source unoccupied. See *Appendix 3—figure 1C* for the illustration of an uninvadabe state constructed by the above procedure in $6C \times 6N \times 36S$ model. Since for each carbon source the above algorithm selects the carbon-limited species (or selects to add no such species) in a unique fashion, there is a single uninvadable state for every stable matching in the Gale-Shapley sense. We are now in a position to predict and enumerate all uninvadable states in our model.

## Lower bound on the number of uninvadable states

To count the number of partitions $(L_1, L_2, ..., L_L)$ such that $\sum L_i = L$, one can use a well known combinatorial method. According to this method, one introduces $L-1$ identical 'separators' (marked with |) which are placed between $L$ identical objects (marked $\cdot$) separating them into $L$ (possibly empty) partitions. For example, for $L = 4$ a partition $0, 1, 0, 3$ would be denoted as $|\cdot||\cdot\cdot\cdot$. The combinatorial number of all possible arrangements of separators and objects is obviously $\binom{2L-1}{L}$. For every such partition the Gale-Shapley theorem guarantees at least one stable matching (that is, at least one uninvadable steady state). The lower bound on the number of uninvadable steady states has to be doubled to account for reversal of roles of carbons and nitrogens. There is a small possibility that we double counted one partition $(1, 1, ..., 1)$. Indeed, the unique uninvadable stable state corresponding to this partition could in principle be counted both when we start from nitrogen sources and when we start from carbon sources. While such uninvadable steady states can and do occur, if only under rather special circumstances described below, they cannot be obtained by the classical Gale and Shapley resident(nitrogen)-proposing algorithm. Indeed, at the last step of the nitrogen-proposing algorithm all carbon sources except just one carbon sources have all their vacancies already filled. Since the algorithm makes all nitrogen sources to go down their preference lists, at this point none of the carbon source whose outgoing arrow has already been accepted would prefer the last nitrogen with a vacancy to the one they have sent an arrow to. Thus, in terms of our adaptation of the Gale-Shapley algorithm to identify all uninvadable states, the last carbon to receive the incoming link from the last nitrogen does not need to be occupied (send a link from C to N) to prevent an invasion, as there is no one willing to invade it because of nitrogen preferences. Thus, in the uninvadable state identified by the Gale and Shapley nitrogen-proposing algorithm, this carbon source would have an incoming arrow but no outgoing arrow. Such states would not be double counted and would not affect our lower bound on the total number of uninvadable states in the model. A partition in which each carbon and each nitrogen source have exactly one vacancy could have more than two (possibly identical) stable states found by the Gale and Shapley nitrogen-proposing and carbon-proposing algorithms could lead to double counting. Hence, in general one may need

to merge a handful of identical states independently found starting from carbon and nitrogen sides.

Then we have $N_{UIS} \geq 2\binom{2L-1}{L} = \binom{2L}{L}$. The Sterling approximation for this expression is $2^{2L}/\sqrt{\pi L}$. Thus the overall lower bound for the number of uninvadable stable states is given by

$$N_{UIS}(L,L) \geq \cdot \binom{2L}{L} \simeq \frac{2^{2L}}{\sqrt{\pi L}} \qquad . \tag{S4}$$

More generally, the number of carbon sources, $K$, is not equal to the number of nitrogen sources, $M$. The resource type with a larger number will always have at least one resource left without input. Thus here one never needs to correct for double counting. Using the same reasoning as for $K = M = L$, the lower bound on the number of resources in this case is given by $\binom{K+M-1}{K-1} + \binom{K+M-1}{M-1} = \binom{K+M}{K}$. Here, the first term counts the uninvadable steady states in which all nitrogen sources are occupied and the partition divides $M$ edges sent by nitrogen sources among $K$ carbon sources, which requires $K - 1$'dividers'. The second term counts the number of uninvadable steady states in which all carbon sources are occupied. Denoting the fraction of carbon resources among all resources as $p = K/(K + M)$ and using the Stirling approximation one gets

$$N_{UIS}(K,M) \geq \cdot \binom{K+M}{K} \simeq$$
$$\simeq \frac{\exp[(K+M)(-p\log p - (1-p)\log(1-p)]}{\sqrt{2\pi(K+M)p(1-p)}} \qquad . \tag{S5}$$

In the case of multiple microbial species using the same pairs of resources, our version of the Gale-Shapley resident-oriented algorithm must be further updated. Let $M$ be the number of nitrogen sources, and $K$ — the number of carbon sources in the ecosystem, $S$ the number of species in our pool, each requiring a pair of resources to grow. As now there may be more than one microbe that uses a given pair of resources $c_i$ and $n_j$, we introduce the notation $B_{ij}^{(r)}$ for the $r$th microbe using the same pair of sources $c_i$ and $n_j$. On average, each nitrogen (carbon) source has $S/K$ ($S/M$) microbes, which are capable of using it. As in the traditional Gale-Shapley algorithm, each nitrogen (carbon) source ranks all microbes capable of using it by their $\lambda^{(n)}$ ($\lambda^{(c)}$).

The way to identify all uninvadable stable states in this case is determined by a variant of the stable marriage problem (or rather the hospital/resident problem) in which every man (and every woman) may have more than one way to propose marriage to the same woman (man). In our model, this corresponds to more than one microbe (a type of marriage) capable of growing on the same pair of carbon (corresponding to, say, men) and nitrogen (corresponding to women) sources. You may think of it as if each participant has several different ways to propose to the person of the opposite sex (send flowers, take to a restaurant, etc). Each of these proposals is ranked by both parties independent of other ways. As far as we know, this variant has not been considered in the literature yet. However, all of the results of the usual stable marriage (or hospital-resident) problem remain unchanged.

One can easily see that our lower bound (*Equation S5*) on the number of uninvadable states (equal to the number of stable marriages in all partitions) remains unchanged. Indeed, it is given by the number of partitions and hence depends only on $K$ and $M$ and not on $S$. However, for $S \gg K \cdot M$ one expects to have many more stable marriages for each partition. Thus the lower bound we have established is likely to severely underestimate the actual number of UIS in the ecosystem.

## Appendix 4

# Conditions of multistability in the 2C × 2N × 4S ecosystem

Below we consider the general case of a 2C × 2N × 4S ecosystem. Let's assume that the selected set of $\lambda$ parameters allow potentially unstable state in which all four species are present. These species form a single loop in the network representation (like state S7 in our 2C × 2N × 4S example). Without loss of generality we may assume that $\lambda-$ parameters satisfy $\lambda_{11}^{(n)} > \lambda_{21}^{(n)}$; $\lambda_{21}^{(c)} > \lambda_{22}^{(c)}$; $\lambda_{22}^{(n)} > \lambda_{12}^{(n)}$; $\lambda_{12}^{(c)} > \lambda_{11}^{(c)}$. The 4-species loop is then formed by the links $C_1 \to N_1$, $N_1 \to C_2$, $C_2 \to N_2$, and $N_2 \to C_2$ In all other cases we may rename C and N resources until the direction of the loop is as stated above. The system also has two uninvadable steady states (A) in which two microbes ($N_1 \to C_1$ and $N_2 \to C_1$) are limited by their nitrogen sources and (B) in which two other microbes are limited by their carbon sources ($C_1 \to N_2$ and $C_2 \to N_1$). These two states have their regions of feasibility in the influx space. In order for these regions to overlap with each other, thereby resulting in bistability within the overlapping region, the yield parameters of species and nutrient supply rates have to satisfy the following conditions.

For the state (A), the conservation laws read as

$$
\begin{aligned}
\frac{\phi_1^{(c)}}{\delta} &= c_1 + \frac{B_{11}}{Y_{11}^{(c)}} \\
\frac{\phi_2^{(c)}}{\delta} &= c_2 + \frac{B_{22}}{Y_{22}^{(c)}} \\
\frac{\phi_1^{(n)}}{\delta} &= \frac{\delta}{\lambda_{11}^{(n)}} + \frac{B_{11}}{Y_{11}^{(n)}} \\
\frac{\phi_2^{(n)}}{\delta} &= \frac{\delta}{\lambda_{22}^{(n)}} + \frac{B_{22}}{Y_{22}^{(n)}}
\end{aligned}
\tag{S6}
$$

The last two relations in (*Equation S6*) define microbe concentrations as $B_{11} = Y_{11}^{(n)} \left( \frac{\phi_1^{(n)}}{\delta} - \frac{\delta}{\lambda_{11}^{(n)}} \right)$, $B_{22} = Y_{22}^{(n)} \left( \frac{\phi_2^{(n)}}{\delta} - \frac{\delta}{\lambda_{22}^{(n)}} \right)$. Substituting these expressions into the first two relations in (*Equation S6*) and invoking the requirements $c_1 > \frac{\delta}{\lambda_{11}^{(c)}}$, $c_2 > \frac{\delta}{\lambda_{22}^{(c)}}$ (guaranteeing that neither of two carbons limits microbes' growth), in the high flux limit we obtain

$$
\begin{aligned}
Y_{11}^{(n)} \phi_1^{(n)} &< Y_{11}^{(c)} \phi_1^{(c)} \\
Y_{22}^{(n)} \phi_2^{(n)} &< Y_{22}^{(c)} \phi_2^{(c)}
\end{aligned}
\tag{S7}
$$

The inequality conditions above are only natural given that the microbes use up their nitrogen fluxes very thoroughly in the state (A) while (at least in the high-flux limit) they not getting even close to consuming all of their carbon supply rates .

The state (B) in the high flux limit will require different conditions, although obtained in a perfectly similar way:

$$
\begin{aligned}
Y_{12}^{(n)} \phi_2^{(n)} &< Y_{12}^{(c)} \phi_1^{(c)} \\
Y_{21}^{(n)} \phi_1^{(n)} &< Y_{21}^{(c)} \phi_2^{(c)}
\end{aligned}
\tag{S8}
$$

Combining *Equation S7* and *Equation S8* one gets

$$\frac{Y_{11}^{(c)}}{Y_{11}^{(n)}} \phi_1^{(c)} > \phi_1^{(n)} > \frac{Y_{21}^{(c)}}{Y_{21}^{(n)}} \phi_2^{(c)}$$

$$\frac{Y_{22}^{(c)}}{Y_{22}^{(n)}} \phi_2^{(c)} > \phi_2^{(n)} > \frac{Y_{12}^{(c)}}{Y_{12}^{(n)}} \phi_1^{(c)} \tag{S9}$$

Hence, for the carbon fluxes ratio, one would have

$$\frac{Y_{21}^{(c)} Y_{11}^{(n)}}{Y_{21}^{(n)} Y_{11}^{(c)}} < \frac{\phi_1^{(c)}}{\phi_2^{(c)}} < \frac{Y_{22}^{(c)} Y_{12}^{(n)}}{Y_{22}^{(n)} Y_{12}^{(c)}} \tag{S10}$$

which implies that for multistability to be possible at least for some ratio of fluxes the yields have to satisfy the following inequality:

$$\frac{Y_{21}^{(c)} Y_{11}^{(n)} Y_{22}^{(n)} Y_{12}^{(c)}}{Y_{21}^{(n)} Y_{11}^{(c)} Y_{22}^{(c)} Y_{12}^{(n)}} < 1 \tag{S11}$$

Note that the **Equation S11** is both necessary and sufficient for bistability between (A) and (B) for some set of supply rates. Indeed, if **Equation S11** is satisfied, $\phi_2^c$ can be chosen arbitrarily (the only thing one would have to mind here is the high-flux limit requirements), then $\phi_1^c$ should be chosen in accordance with **Equation S10**, and any nitrogen fluxes satisfying **Equation S9**. All the procedures are legitimate whenever **Equation S11** holds. Once chosen in the way described above, the point $(\phi_1^{(c)}, \phi_2^{(c)}, \phi_1^{(n)}, \phi_2^{(n)})$ of the influx space will make both (A) and (B) steady states feasible.

In a more general case of an arbitrary number of nutrients of each type, the conditions allowing for bistability or even multistability can be expressed by simple inequalities connecting yields and fluxes in combinations dictated by network topology of potentially bistable states in a very similar way to the simple case presented above. Thus the solution to the puzzle of why roughly half of all possible yield combinations has no multistability whatsoever becomes intuitively clear. Indeed, these yields and fluxes must come in 'dimensionless' combinations so that any inequality can be written as a function of only C:N stoichiometry $S_{ij}^{(C:N)} = Y_{ij}^{(n)} Y_{ij}^{(c)}$ for all microbial species present in any of the set of potentially multistable states. Note that should nitrogen and carbon yields exchange places for each of these microbes, the key inequality similar to **Equation S11** would be reversed, thus prohibiting multistability where it was permitted and vice versa.

In the yield space, the proposed swap of carbon and nitrogen yields of all species $Y^{(c)} \rightarrow Y^{(n)}$, $Y^{(n)} \rightarrow Y^{(c)}$ is a volume preserving transformation. This means that, for each set of potentially multistable states, the fraction of the yield space favouring multistability is exactly the same as the fraction prohibiting multistability. In a possible (albeit unlikely) scenario when the suggested permutation affects not only the potential multistability in question, but also some other multistable conditions, 'turning off' one multistability might in some cases 'turn on' others. This is why the ultimate empirical probability of multistability for a given combination of species' yields might somewhat deviate from 1/2.

## Appendix 5

### Proof of the absence of multistability when all species have identical nutrient stoichiometries

For randomly selected nutrient supply rates, the chance for the system to end up in a given steady state is proportional to the volume of its region of feasibility. In a $(K + M)-$ dimensional space, where each point corresponds to the set of K carbon and M nitrogen nutrient supply rates, regions of feasibility for some uninvadable states may intersect. In this case, the area they have in common defines the initial conditions at which multistability can occur. Naturally, for the effect to be observable, that shared area has to be $(K + M)-$ dimensional, else its volume and, correspondingly, the chance to encounter such a situation in a real ecosystem would be negligibly small.

In the general case, when the microbes differ in terms of their stoichiometry, the steady state conditions dictate, for carbon supply rates:

$$\frac{\phi_i^{(c)}}{\delta} = \sum_k \frac{B_{ik}}{Y_{ik}^{(c)}} + c_i \tag{S12}$$

where $\frac{\phi_i^{(c)}}{\delta}$ is the abiotic value of the concentration of the nutrient $c_i$, $i$ ranging from 1 to $K$, $B_{ik}$, as usual, is the concentration of the microbe growing on $c_i$ and $n_k$ (equal to zero if the microbe is absent in the state), $Y_{ik}^{(c)}$ — the yield of the microbe with respect to its carbon source.

Similarly, for nitrogen supply rates one would have:

$$\frac{\phi_j^{(n)}}{\delta} = \sum_l \frac{B_{lj}}{Y_{lj}^{(n)}} + n_j \tag{S13}$$

here $\phi_j^{(n)}$ being the abiotic value of the concentration of the nutrient $n_j$, $Y_{lj}^{(n)}$ – the yield of the microbe using $c_l$ and $n_j$ with respect to its nitrogen source, $j$ ranging from 1 to $M$.

When multistability occurs, there exist at least two different sets of values of $B_{ij}$, $c_l$ and $n_m$ such that either of them can fit both *Equation S12* and *Equation S13* for the same values of nutrient supply rates. After marking one of them with the symbol ~ for carbon sources one gets:

$$\sum_k \frac{B_{ik}}{Y_{ik}^{(c)}} + c_i = \sum_m \frac{\tilde{B}_{im}}{Y_{im}^{(c)}} + \tilde{c}_i \tag{S14}$$

and for nitrogen sources:

$$\sum_l \frac{B_{lj}}{Y_{lj}^{(n)}} + n_j = \sum_p \frac{\tilde{B}_{pj}}{Y_{pj}^{(n)}} + \tilde{n}_j \tag{S15}$$

In the general case, where the microbes differ in their yields, *Equation S14* and *Equation S15* taken together for all relevant $i, j$, describe a $(K + M)$–dimensional area. Indeed, the nutrient concentration on any occupied resource is fixed to $\delta/\lambda_\alpha^{(c,n)}$, where $\alpha$ is the species limited by this resource. Conversely, for any unoccupied resource nutrient concentration is free to vary as long as it is larger than $\delta/\lambda_\beta^{(c,n)}$, where $\beta$ is the species using this resource in a given state which has the smallest value of $\lambda^{(c,n)}$. In other words each unoccupied resource contributes one degree of freedom, while occupied resources do not. There is also a positive microbe concentration value for any occupied nutrient source. Thus, the total number of the unknowns is being equal to $K + M$ on the left side and $K + M$ on the right side of the system of the equations. The total number of the unknowns is thus $2K + 2M$. The number of restrictions provided by *Equation S14* and *Equation S15* being $K + M$, the area defined by the system of the equations turns out to be $(K + M)-$ dimensional.

If the stoichiometry (yields) of all species are equal to each other, without loss of generality one can assume them all to be equal to unity. Indeed, in this case the units of resource concentrations or bacterial abundances can be universally changed to match each other so that one unit of resource is used to make exactly one unit of bacterial abundance.

For identical stoichiometry of species there is an additional constraint on our $2K + 2M$ variables. Indeed, after adding up all equations *Equation S14* and then, separately, all equations *Equation S15*, and comparing the results of these additions we get

$$\sum_i (\frac{\phi_i^{(c)}}{\delta} - c_i) = \sum_j (\frac{\phi_j^{(n)}}{\delta} - n_j) \quad . \tag{S16}$$

This reflects the fact that for identical stoichiometries both sides of *Equation S16* are equal to the total biomass concentration $\sum_{ij} B_{ij}$ of all surviving microbes. Hence, if the two sets of concentrations satisfying both *Equation S14* and *Equation S15* exist, under the degenerate stoichiometry condition they must also satisfy an additional constraint

$$\sum_l c_l - \sum_m n_m = \sum_k \tilde{c}_k - \sum_p \tilde{n}_p \tag{S17}$$

as the difference between the total concentration of carbon nutrients and that of nitrogen nutrients must be the same in both states by the virtue of *Equation S16*.

The additional constraint *Equation S17* reduces the dimensionality of the multistability area to $K + M - 1$, so in the $(K + M)-$ dimensional space of nutrient concentrations it has zero volume.

