## [Decision Letter]

Thank you for submitting your article "Multistability and regime shifts in microbial communities explained by competition for essential nutrients" for consideration by *eLife*. Your article has been reviewed by three peer reviewers, including Wenying Shou as the Reviewing Editor and Reviewer #1, and the evaluation has been overseen by Ian Baldwin as the Senior Editor.

The reviewers have discussed the reviews with one another and the Reviewing Editor has drafted this decision to help you prepare a revised submission.

Summary

"Multistability and regime shifts in microbial communities explained by competition for essential nutrients" by Dubinkina et al. explores the presence of diverse, multiple stable states for diverse ecosystems. The model builds on the groups earlier work using results from game theory on the stable marriage problem to ask under what conditions it is possible to have multistability in diverse communities. The authors employ an idealized model of competition for essential nutrients to study patterns of multistability and diversity in microbial ecosystems. By computationally enumerating steady states and classifying them by stability, the authors establish that multistability is a typical feature of resource competition between specialized consumers. Specifically, they show that multistability emerges robustly when nutrient supplies are balanced and competitors have different C:N stoichiometries. The topic of how diversity is maintained in microbial ecosystems is likely to be of broad interest to the readership of *eLife*. While the current model is quite simplified compared to any real multispecies community, the authors do a good job of highlighting general lessons from the model that make testable predictions for natural and experimental systems. Overall, all three reviewers found the work interesting.

Essential revisions

Essential revisions fall under three categories:

1) Present your model more clearly;

2) Add mechanisms and intuitions (rather than simply describing your data);

3) Discuss generalizability of the results, including sensitivity to parameter choices.

To help you revise, we attach the reviews.

Reviewer #1:

My main suggestions are as following:

1) The authors define states by the presence/absence of species, and if present, which nutrient it would be limited in. Out of the theoretical possible states, the authors found all steady states by noting a correspondence of their problem and the stable marriage problem. This correspondence needs to be explained in an accessible form (perhaps in a figure).

2) Put more emphasis on going through the data and explain what they mean. For example, in Figure 1B: why do we get these states? For the two-species 11, 22 cases, one could imagine a different state where 11 is limited for carbon, and 22 is limited for nitrogen. Why is this case not seen? Does this have something to do with your parameter choices? For Figure 3B, why does the likelihood of multistability peak at C:N supply rates similar to the average C:N stoichiometry? If you change the average C:N stoichiometry ratio by 10-fold, does the peak change by 10-fold? Seems that the authors were simply describing results rather than also giving intuition on why the results arose.

3) Subsection “Three criteria for stability of microbial communities” paragraph four: not clear why invasion does not depend on nutrient supply rates, given that nutrient supply rates could change growth rates.

4) Abstract: Multistability requires different stoichiometries of essential nutrients. How different does "different" have to be? It would be good to supplement with previous experimental data on these ratios and discuss.

Reviewer #2:

My main concern is really about how much the conclusions from this particular model used by the authors generalize to a more general setting. I think the assumptions of the model are quite stringent (and need to be in order to really use the technical apparatus of game theory/stable marriage problem they use):

Each species can use only one C and one N resource; a single metabolite cannot provide both C and N (even though we know for example bacteria can and do use amino acids as C sources).

As the authors point out, this gives rise to non-generic behavior such as the fact:

"Unlike other consumer resource models, in our model the dynamic stability of a state with respect to species invasions does not depend on nutrient supply rates."

At the minimum, the authors have to argue what depends on these assumptions and what does not. Comparing the results to more standard Consumer Resource Models (CRMs) would go a long way at clarifying this. Despite this criticism, I want to emphasize that I think the paper explores and interesting question and gives rise to interesting hypothesis that are interesting and potentially testable.

I now give more detailed comments:

Technical Comments:

The section on Monte-Carlo sampling is extremely unclear. What exactly is the distribution that is sampled? Is this done only in the high-flux regime? What is assumed about the R matrix being invertible? Is this somehow an assumption that the Y are "generic"? It is very unclear to me why one cannot be feasible on a low-dimensional manifold. This is usually a generic situation in most CRMs so why is it excluded here. I would appreciate a much more technical discussion, perhaps with pseudo code so that I can better understand if and when the arguments generalize.

The equations in subsection “Conditions of multistability in the 2C x 2N x 4S ecosystem” have typos (extra factors of Y).

I did not understand in the section on lower-bound, why "double counting" can be discarded (either when # of C = # of N or otherwise).

It is very unclear how sensitive are the results to different "draws" of the λ and Y. There is a particular realization given in SI Tables but it would be nice to have a sense of the fluctuations rather than just for one network.

Another general way of modeling essential resources is to use a function of the fromg=(∑αciα/Rα)−1where the cIα are constants with the same units as Rα that encode ratios of resources required for building biomass (see Taillefumier et al., 2017). Recently, it was shown in arxiv:1901.09673 that such functions can also give rise to multi-stability using a Minimum Environmental Perturbation Principle based on measuring distances from the input flux vectors with appropriate distance metric. This suggests it should be possible to generically construct input fluxes into the system with multiple steady-states by finding input vectors that are equally distant from the intersection of all ZGNIs for a different, more traditional class of CRMs where species can eat many resources. This again raises questions about what generalizes from this simple but interesting model. It seems that this would be more in line with the generalization in equation in subsection “Extensions of the model” but it may qualitatively change the results.

Conceptual Comments:

I really enjoyed the idea of how by continuously changing the environments one cannot connect very different stable states. This reminds me of old arguments in evolutionary theory on neutralness in adaptation by Fontana and collaborators (PNAS 1996) where very different evolutionary minima could be connected even though they changed differently. This is just a thought/comment not a suggestion for investigation.

The idea that equalizing resources promotes diversity is now something that has been found in various CRMs (Posfai et al., 2017; Tikhonov et al; Marsland et al., 2019). The idea is that one gets neutral like dynamics. Is there a way to reconcile these ideas with the multi-stability? It is curious that both these dynamics like equalizing resources and I would like to understand what, if any, the authors think the connection is.

I am really confused as stated above how these results depend on the available resource species pool. In particular, what if I changed the statistics of things were generated, yields, etc.? I think it would be nice to at least understand these things at least semi-quantitatively.

Reviewer #3:

1) Existing literature on competition for essential nutrients.

The authors state that "while real-life ecosystems driven by competition for multiple essential nutrients have been studied experimentally Fanin et al., 2016; Browning et al., 2017; Camenzind et al., 2018, the resource-explicit models capturing this type of growth are not so well developed beyond the foundational work by Tilman, 1982." Although this is generally true, it would be helpful to reference the work that does exist, to orient the reader as to whether the current results are surprising. Huisman's paper (Nature 402 1999) is relevant, as is the plankton literature more generally.

2) Assumptions and parameters of the model.

The assumption of complete specialization to a single carbon and nitrogen resource is very limiting. More care should be taken to justify it in the context of real ecosystems. Would the results hold qualitatively if this assumption were relaxed? Why or why not? Furthermore, each model seems to feature only one realization of the growth coefficient λs. The authors don't need to redo everything for new parameters, but they should confirm that their results are not specific to these parameters. Finally, in the current model a species' λs and Ys (growth yields) are uncorrelated. However, it would be more biologically plausible if these were subject to trade-offs for each species (e.g. high growth rates correspond to low yields). It would be informative if the authors would check the case where λY=constant to confirm that their results do not depend on λ and Y being uncorrelated.

3) Explanation of Figure 2E-F

The authors should try to provide an intuitive, mechanistic understanding of these computational results. Why does state 2 have the largest feasibility volume, and why are regime shifts possible between some states and not others? If I understood the simple case of 2e, I would have a better appreciation of 2f. A discussion of the ecological implications would also be welcome here.

4) 1D manifold in Figure 3C

In the same spirit as point #3, the authors should try to explain this result and its implications. This is certainly an interesting observation, but as of now it's asserted without any mechanistic understanding.

5) Competitive rank and diversity

The "competitive rank" metric is unintuitive, at least as presented. The authors should explain more clearly why ranks run from 1-6 and why they're a species property independent of the environment. The surprising result that a rank 5.5 species is uninvadable 20% of the time should be explained. What is a specific scenario in which this occurs, and why?

6) Volume in abundance space

The authors focus on the feasibility volume of a state in the space of nutrients. However, it is also natural to consider the volume of the basins of attraction in the space of species abundances for a particular state in a given environment, when multiple stable states are possible. Does the volume of the basin of attraction for a state relate to its feasibility volume?

---

## [Author Response]

Reviewer #1:My main suggestions are as following:1) The authors define states by the presence/absence of species, and if present, which nutrient it would be limited in. Out of the theoretical possible states, the authors found all steady states by noting a correspondence of their problem and the stable marriage problem. This correspondence needs to be explained in an accessible form (perhaps in a figure).

We thank the reviewer for this suggestion. To address this we added a paragraph with additional details clarifying the general ideas behind the correspondence between our model and the stable marriage problem (see subsection “Microbial community growing on two types of essential nutrients represented by

85 multiple metabolites” paragraph six and “Three criteria for stability of microbial communities” paragraph two). These are further explained in the Materials and methods section "Identification of all states and classification of them as invadable or uninvadable”. We also believe that knowing the exact correspondence to the stable marriage problem is not essential for understanding of our main results and may even be distracting to our readers.

We have thought long and hard if this correspondence could be illustrated in an additional figure in the main text of the article and (after long deliberations) decided to leave it for supplementary materials (see Appendix 3-figure 1). Unlike a straightforward mapping to the stable marriage problem outlined in our previous paper (Goyal et al., 2018), the relation to the stable marriage problem described in this study, while mathematically exact, is a bit more formal. Indeed, as explained in the section "Stable matching approach for identification and classification of steady states" of the supplementary notes, "marriages" in the C:N model are unidirectional and represented by directed arrows between C and N sources. It would be confusing to readers to work with two different graphical representations of the same problem (compare Figure 1A and Appendix 3-figure 1 in the current version of the manuscript).

2) Put more emphasis on going through the data and explain what they mean. For example, in Figure 1B: why do we get these states? For the two-species 11, 22 case, one could imagine a different state where 11 is limited for carbon, and 22 is limited for nitrogen. Why is this case not seen? Does this have something to do with your parameter choices?

We thank the reviewer for bringing this up and helping us to clarify our results. Indeed, the list of uninvadable states illustrated in Figure 1B is determined by our specific choice of model parameters λ quantifying competitiveness of individual microbial species (see Supplementary file 1). For this specific case, the state where the species 11 is limited by carbon, and the species 22 – by nitrogen is always invadable by the species 21 and hence not listed among uninvadable states in Figure 1B. We added the explanation of another case to the main text to provide readers with a specific example and additional understanding of how the relative ranking of lambdas defines the invadability of a state (paragraph two subsection “Three criteria for stability of microbial communities”).

For Figure 3B, why does the likelihood of multistability peak at C:N supply rates similar to the average C:N stoichiometry? If you change the average C:N stoichiometry ratio by 10-fold, does the peak change by 10-fold? Seems that the authors were simply describing results rather than also giving intuition on why the results arose.

The short answer is yes: for a 10:1 stoichiometry the multistability peak in Figure 3B will shift to a 10-fold higher ϕ(c)/ϕ(n) ratio. In general, the C:N stoichiometry averaged across all species can be renormalized to the 1:1 ratio used in our study by redefining the nutrient concentrations and supply rates. For example if the average stoichiometry is 10:1 one can measure all carbon concentrations c_i_ in 10-fold larger units than nitrogen concentrations n_j_ thereby renormalizing the stoichiometry to 1:1. We have added a brief explanation of this in paragraph three of subsection “Patterns of multistability”

3) Subsection “Three criteria for stability of microbial communities” paragraph four: not clear why invasion does not depend on nutrient supply rates, given that nutrient supply rates could change growth rates.

Indeed, nutrient supply rates change nutrient concentrations, which in turn, change the growth rates of species. However, they do so proportionally for all species using the same nutrient. Throughout the manuscript and in all our numerical simulations we are working in the high nutrient supply regime. In this regime, all individual microbes would be able to grow as isolates. Hence, when multiple species are introduced, their survival is solely due to the competition between them. In this case, the invadability of a state is fully determined by the rank-order of lambdas of the invading species and all species present in the state. In the opposite regime, where at least some of the supply rates are low the set of uninvadable states may expand beyond what we predict. However, our uninvadable states would remain uninvadable even in the low nutrient supply regime.

To clarify this argument we have rephrased the text. It now reads:

"Note that, in the regime of our model, where the supply of all nutrients is high, i.e. ϕ(c,n)≫δ2/λα(c,n), invadability of individual states does not depend on supply rates. Indeed, in this regime the outcome of an attempted invasion is fully determined by the competition between species, which in turn depends only on the rank-order of λ's of the invading species relative to the species currently present in the ecosystem“. We also clarified it in the respective subsection in Materials and methods.

4) Abstract: Multistability requires different stoichiometries of essential nutrients. How different does "different" have to be? It would be good to supplement with previous experimental data on these ratios and discuss.

Multistability is possible for arbitrarily small variation of stoichiometry, but in this case, it would be confined to a very narrow region of nutrient supply rates. We have previously illustrated it in what was Figure S3 of the original submission. Inspired by this comment and comments of other reviewers we decided to move this figure to the main text of the manuscript (see Figure 5 in the revised submission). It shows that the smaller is the variance of stoichiometries (the x-axis) – the less likely one is to encounter multistability for randomly selected combination of nutrient supply rates (the y-axis of Figure 5). We have added a discussion of this along with comparison with experimental data in subsection “Multistability requires diverse species stoichiometry”.

We also showed where the real-life soil microbial ecosystem stoichiometry variation lies on this plot.

We also added the experimentally determined stoichiometry variance for real ecosystems (mammalian gut, phytoplankton, and soil). This suggests that multistability might not be very rare (up to 10% likelihood for soil). We also wrote supplementary note (Appendix 5) that proves that multistability is impossible if all species have identical nutrient stoichiometries.

Reviewer #2:1) My main concern is really about how much the conclusions from this particular model used by the authors generalize to a more general setting. I think the assumptions of the model are quite stringent (and need to be in order to really use the technical apparatus of game theory/stable marriage problem they use):Each species can use only one C and one N resource; a single metabolite cannot provide both C and N (even though we know for example bacteria can and do use amino acids as C sources).As the authors point out, this gives rise to non-generic behavior such as the fact:"Unlike other consumer resource models, in our model the dynamic stability of a state with respect to species invasions does not depend on nutrient supply rates."At the minimum, the authors have to argue what depends on these assumptions and what does not. Comparing the results to more standard Consumer Resource Models (CRMs) would go a long way at clarifying this. Despite this criticism, I want to emphasize that I think the paper explores and interesting question and gives rise to interesting hypothesis that are interesting and potentially testable.

We agree with the reviewer that the assumptions of our model are both stringent and crucial for the exact mapping to the Stable Marriage Problem. However, the conceptual framework in which we reduce multistability between a pair of states to a problem of overlaps of their feasibility regions is general and applicable to a broad variety of Consumer Resource Models. The extent to which our findings hold in other CRM models is the subject of ongoing research in our lab. We added a paragraph comparing one of the key results in our model to that in other CRMs (subsection “Multistability requires diverse species stoichiometry”). We also emphasized the universality (or lack thereof) of our results in other CRMs in subsection “Interplay between diversity and stability in ecosystems with multiple essential nutrients”. Finally, we cited arxiv:1901.09673, which has a number of additional insights into multistability in CRMs.

In the end of our manuscript we propose two generalizations of our model that can be investigated numerically: the one in which g(C,N)=min(∑iλi(c)Ci,∑iλj(c)Nj) (a hybrid of our model with the MacArthur model) and another one in which microbes diauxically shift between resources (a hybrid of our model with our previously published model (Goyal et al., 2018) also mapped onto the Stable Marriage Problem). We expect our mathematical techniques to be at least partially extendable to both of these models.

I now give more detailed comments:Technical Comments:2) The section on Monte-Carlo sampling is extremely unclear. What exactly is the distribution that is sampled? Is this done only in the high-flux regime?

We Monte-Carlo (MC) sampled our flux space randomly (and uniformly in linear space) between 10 and 1000. For each supply combination point, all nutrients were sampled independently as described in Materials and methods. The lower bound was selected in such a way that all our MC sampling was done for the high nutrient supply regime (indeed our λ’s are >10 and δ=1 giving the lower bound of the high flux regime <0.1 well below the lowest nutrient flux of 10. It is now described in more detail in the Materials and methods section.

3) What is assumed about the R matrix being invertible? Is this somehow an assumption that the Y are "generic"?

Yes, to instantiate our model we randomly generated Y, which all but guaranteed that the matrices Rp in Equations 5,6 are invertible. Thus given nutrient supply rates we can always calculate bacterial abundances and concentrations of all nutrients, which are not limiting to any microbes. The problem is made non-trivial by the feasibility constraint that all of these have to be positive.

4) It is very unclear to me why one cannot be feasible on a low-dimensional manifold. This is usually a generic situation in most CRMs so why is it excluded here. I would appreciate a much more technical discussion, perhaps with pseudo code so that I can better understand if and when the arguments generalize.

One common strategy in CRMs is to work with nutrient supply rates artificially constrained to a low-dimensional manifold (e.g. by adding a constraint that ∑iϕi=ϕtotal). Such constraints would not change our results except by reducing all of the dimensions used in our study by 1. For example, it would turn K^+^M dimensional space of ϕ into K^+^M-1 dimensional space of constrained ϕ. At the same time the dimensionality of the boundaries between states would go down from K^+^M-1 to K^+^M-2. Hence, a shared boundary between states still would not imply multistability.

Another argument on why we ignore states feasible only on a lower-dimensional manifold is as follows.

We evaluate the feasibility of a state separately for each set of nutrient supply rates ϕ. If a state is feasible only on a low-dimensional subset of ϕ, any small perturbation of these will take the system out of this state. Such a state would be nearly impossible to detect in our Monte Carlo simulations. It is also unlikely to be experimentally observed in real microbial ecosystems unless there is a special reason why nutrient supply rates would satisfy this constraint. This treatment of feasibility volumes is standard (see e.g. Grilli et al., 2016 for GLV model or Butler and O'Dwyer, 2018 for the MacArthur model).

The equations in subsection “Conditions of multistability in the 2C x 2N x 4S ecosystem” have typos (extra factors of Y).

These have been corrected.

5) I did not understand in the section on lower-bound, why "double counting" can be discarded (either when # of C = # of N or otherwise).

We tried to better explain our proof that double counting is impossible for stable states identified by the Gale-Shapley men (nitrogen) -proposing algorithm. These are the only states we use in our calculations of the lower bound on the number of uninvadable states. The modified explanation is in Appendix 3.

6) It is very unclear how sensitive are the results to different "draws" of the λ and Y. There is a particular realization given in SI Tables but it would be nice to have a sense of the fluctuations rather than just for one network.

We agree with the reviewer that the question of how parameters λ and Y affect the set of states in our model is an interesting topic for future investigation. During our preparation of this manuscript, we experimented with different sets of lambdas and Y's. We noticed that while the exact set of dynamically stable and uninvadable states depends on our choice of parameters, their number and statistical properties remain the same. We also noticed that the choice of Y’s has a much stronger influence on multistability, which is the main focus of this study. This influence was illustrated in what was the Figure 3S (now moved as Figure 5 in the main text) with different draws of Y and the same set of λ's in the 2C x 2N x 4S model. Similar parameter sensitivity analysis of the 6C x 6N x 36S model was prohibitively computationally expensive.

7) Another general way of modeling essential resources is to use a function of the fromg=(∑αciα/Rα)−1 where the cIα are constants with the same units as Rα that encode ratios of resources required for building biomass (see Taillefumier et al., 2017). Recently, it was shown in arxiv:1901.09673 that such functions can also give rise to multi-stability using a Minimum Environmental Perturbation Principle based on measuring distances from the input flux vectors with appropriate distance metric. This suggests it should be possible to generically construct input fluxes into the system with multiple steady-states by finding input vectors that are equally distant from the intersection of all ZGNIs for a different, more traditional class of CRMs where species can eat many resources. This again raises questions about what generalizes from this simple but interesting model. It seems that this would be more in line with the generalization in equation in subsection “Extensions of the model” but it may qualitatively change the results.

We have now cited arxiv:1901.09673 and the Minimum Environmental Perturbation Principle (MEMP), which may provide new insights on general question of multistability in CRMs. We do not fully understand the connection between our methods and MEMP. For example, our model does not have multistability if the yields of all microbes on a given nutrient are equal to each other (see proof in Appendix 5). That is to say, in this case the feasibility regions of different uninvadable states do not overlap with each other, except on a low-dimensional manifold of nutrient supply rates. The model from arxiv:1901.09673 with g=(∑αciα/Rα)−1 can demonstrably be multistable (see their Figure 5C). However, it is unclear whether this multistability is limited to a low-dimensional manifold composed of "input vectors that are equally distant from the intersection of all ZGNIs". If it does, our results agree with arxiv:1901.09673. Overall, we believe that these questions could be fruitfully explored in future extensions of our model and are out of scope for this study.

Conceptual Comments:8) I really enjoyed the idea of how by continuously changing the environments one cannot connect very different stable states. This reminds me of old arguments in evolutionary theory on neutralness in adaptation by Fontana and collaborators (PNAS 1996) where very different evolutionary minima could be connected even though they changed differently. This is just a thought/comment not a suggestion for investigation.

We thank the reviewer for this positive feedback. We thought about the analogy between addition of species in our model and addition of mutations allowed by natural selection in models similar to Fontana, 1996. We explained this observation in more detail in our manuscript (see subsection “Control of microbial ecosystems exhibiting multistability and regime shifts”). We also have partially implemented this analogy in a forthcoming manuscript where we explore possible changes in the community state along different trajectories in nutrient supply rates and species additions.

9) The idea that equalizing resources promotes diversity is now something that has been found in various CRMs (Posfai et al., 2017; Tikhonov et al; Marsland et al., 2019). The idea is that one gets neutral like dynamics. Is there a way to reconcile these ideas with the multi-stability? It is curious that both these dynamics like equalizing resources and I would like to understand what, if any, the authors think the connection is.

We have rather good conceptual understanding of this issue. In our model, multistability and species diversity are two sides of the same coin. In the region where nutrient supplies are equalized (well-balanced region in our notation), there is an excess of states with high diversity of species (see our Figure 4A). Some of these states are dynamically stable – in this case, high diversity is maintained. Dynamically unstable states on the other hand break up into bistable pairs of lower-diversity states.

It is easy to understand this relation using the example of Tilman model with a single C and a single N resource. In this case, the well-balanced region is located near the diagonal of the C-N plane. For such nutrient supplies one has either the coexistence between two species (a high-diversity state) or bistability between two low-diversity ones in which either one or the other species survive. The choice between these two scenarios is dictated by the combination of yields and competitiveness of two species, which determine dynamical stability of the two-species steady state. Similar arguments apply to our model with many C and many N resources. However, since the number of states that could be dynamically unstable is large, there is no mutual exclusivity between multistability and high species diversity in the well-balanced region. We have cited this connection between our model and other CRMs in subsection “Interplay between diversity and stability in ecosystems with multiple essential

nutrients”.

10) I am really confused as stated above how these results depend on the available resource species pool. In particular, what if I changed the statistics of things were generated, yields, etc.? I think it would be nice to at least understand these things at least semi-quantitatively.

We plan to explore some of these topics in future papers. Mapping to the Stable Marriage Problem proves that the statistics (meaning PDFs) of lambdas does not matter for our model. The relative ranks of lambdas fully determine the set of steady states in our model and their invadability. However, as discussed in the paper and shown in Figure 5 (formerly Figure S3) the standard deviation of yields has a strong impact on the extent of multistability in our model. Moreover, we suspect that tradeoffs or correlations between λ^(c)^ and λ^(n)^ or λ’s and yields might affect the number of stable states in the model. Indeed, we have demonstrated this in our previous work (Goyal et al., 2018) in a different model also based on the stable marriage problem. We now speculate about potential consequences of different tradeoffs in a dedicated section of the Discussion in subsection “Multistability and the total number of states are affected by tradeoffs”.

Reviewer #3:1) Existing literature on competition for essential nutrients.The authors state that "while real-life ecosystems driven by competition for multiple essential nutrients have been studied experimentally Fanin et al., 2016; Browning et al., 2017; Camenzind et al., 2018, the resource-explicit models capturing this type of growth are not so well developed beyond the foundational work by Tilman, 1982." Although this is generally true, it would be helpful to reference the work that does exist, to orient the reader as to whether the current results are surprising. Huisman's paper (Nature 402 1999) is relevant, as is the plankton literature more generally.

We expanded our literature list with several experimental papers quantifying the variability of nutrient stoichiometry in several real ecosystems (phytoplankton in oceans, microbes in soil, and mammalian gut). These references helped us to put a number on likelihood of multistability in each of these systems, which is now shown in Figure 5 (see subsection “Multistability requires diverse species stoichiometry” for the discussion of this figure). We also added an important reference to the Huisman's paper, which we previously overlooked (see paragraph two of the Introduction).

2) Assumptions and parameters of the model.The assumption of complete specialization to a single carbon and nitrogen resource is very limiting. More care should be taken to justify it in the context of real ecosystems. Would the results hold qualitatively if this assumption were relaxed? Why or why not?

The mathematical mapping to the stable marriage problem depends on species' complete specialization to use a single carbon and a single nitrogen resource. However, the conceptual framework in which the multistability between uninvadable states is connected to overlaps of their feasibility regions of nutrient supply rates is completely general. We expect the following findings of our model to hold in a broad variety of consumer resource models and real ecosystems: 1) positive correlation between the likelihood of finding multistable nutrient supply combinations and the variability of nutrient stoichiometry (Figure 5 and discussion in subsection “Multistability requires diverse species stoichiometry”); 2) positive correlation between multistability (Figure 3C), species diversity (Figure 4A) and balanced nutrient supply (see the discussion in response to point 9 by reviewer #2 and subsection “Interplay between diversity and stability in ecosystems with multiple essential nutrients”). The extent to which these and other findings can be extended to other CRM models is the subject of ongoing research in our lab as outlined in Discussion.

Furthermore, each model seems to feature only one realization of the growth coefficient λs. The authors don't need to redo everything for new parameters, but they should confirm that their results are not specific to these parameters.

Mapping to the Stable Marriage Problem proves that the statistics of lambdas does not at all matter for our model. The relative ranks of lambdas fully determine the set of steady states in our model and their invadability in the high nutrient supply regime (see subsection “Three criteria for stability of microbial communities” for explanation). We suspect that tradeoffs or correlations between different lambdas might affect the number of stable states in the model (as discussed in subsection “Multistability and the total number of states are affected by tradeoffs”). Indeed, we have demonstrated this in our previous work (Goyal et al., 2018) in a different model also based on stable marriage problem.

During the preparation of our manuscript we experimented with different sets of λ's and Y's. We noticed that while the exact set of dynamically stable and uninvadable states depends on our choice of parameters, their statistical properties remain the same. We also noticed that the choice of Y’s has a much stronger influence on multistability, which is the main focus of this study. This influence was illustrated in what was the Figure 3S (now moved as Figure 5 in the main text) with different draws of Y and the same set of λ's in the 2C x 2N x 4S model (see subsection “Multistability requires diverse species stoichiometry” for the discussion of this figure). Similar parameter sensitivity analysis of the 6C x 6N x 36S model was prohibitively computationally expensive.

Finally, in the current model a species' λs and Ys (growth yields) are uncorrelated. However, it would be more biologically plausible if these were subject to trade-offs for each species (e.g. high growth rates correspond to low yields). It would be informative if the authors would check the case where λY=constant to confirm that their results do not depend on λ and Y being uncorrelated.

We thank the reviewer for this comment as it lead us to an important discussion. While we do not study any correlated sets of parameters in the current variant of our model, we added a subsection in the Discussion section that summarizes our ideas on this topic (see subsection “Multistability could be affected by tradeoffs”).

3) Explanation of Figure 2E-FThe authors should try to provide an intuitive, mechanistic understanding of these computational results. Why does state 2 have the largest feasibility volume, and why are regime shifts possible between some states and not others? If I understood the simple case of 2e, I would have a better appreciation of 2f. A discussion of the ecological implications would also be welcome here.

We thank the reviewer for this comment, as it helped us to improve the manuscript. The states #1 and #2 have the lowest species diversity (2 species as opposed to 3 species in states #3-6), thus they can be realized for a larger number of nutrient supply combinations (this diversity/feasibility dependence is illustrated in Figure 4C and discussed in the manuscript). The fact that the volume of the state #2 is larger than that of the state #1 is purely due to chance and is determined by our specific parameter choice.

We also added a new type of edges (marked blue) to Figure 2E, which mark continuous, smooth transitions between states without discontinuous regime shifts. These appear due to shared boundaries (instead of overlaps) between the pairs of states. Using the network shown in Figure 2E we developed some mechanistic intuition behind which pairs of states are connected by a bistable regime shifts, which have a shared boundary, and which have no direct transition at all. As one can see, the driver of all bistable switches shown in Figure 2E are the mutual exclusivity of pairs of keystone species (1,1), (2,2) and (1, 2), (2,1) correspondingly. Once one of these pairs is established, it cannot be invaded by another pair. The state #7 with all 4 species present is dynamically unstable and for some nutrient supply rates it spontaneously collapses to one of two states with different keystone species giving rise to bistability. Conversely, a shared boundary is possible for some of the states sharing the pair of keystone species.

Using the example of our 2C x 2N x 4S model, we can now conjecture the following intuitive view of regime shifts in more complex models including our 6C x6N x 36S model. Every ecosystem has a relatively small number of "drivers of bistability’’ generated by certain dynamically unstable states (like state #7). Each of these states can collapse into mutually exclusive pairs of stable states each with its own set of "keystone" species. In addition to keystone species the states could also include ‘’satellite’’ species that do not generally affect the bistable switch between states.

Our detailed explanations of these effects were added to the manuscript (see Results and Discussion sections).

4) 1D manifold in Figure 3CIn the same spirit as point #3, the authors should try to explain this result and its implications. This is certainly an interesting observation, but as of now it's asserted without any mechanistic understanding.

We currently do not have a mechanistic interpretation of these results. We suspect that it is a consequence of some special type of Lyapunov function in our problem (see the Discussion) but we have not found this function and thus cannot answer this question. We added the following text describing possible ecological implications of our results in real microbial ecosystems:

"It is tempting to conjecture that some variant of our model may explain similar arrangements of clusters of microbial abundances commonly seen in PCA plots of real ecosystems. If this is the case, the gaps between neighboring clusters would correspond to dynamically unstable states of the ecosystem, which may be experimentally observable as long transients in species composition."

5) Competitive rank and diversityThe "competitive rank" metric is unintuitive, at least as presented. The authors should explain more clearly why ranks run from 1-6 and why they're a species property independent of the environment.

We clarified our definition of the species competitiveness rank in the manuscript (see subsection “Patterns of diversity and structural stability of states”). The competitive rank of a species for carbon (nitrogen) is given by the rank order of its λ^(c)^ (λ^(n)^) among all species using the same carbon (nitrogen) source. Since in our example there are L=6 species using each carbon (nitrogen) resource, the rank varies between 1 for the most competitive species in our pool and 6 for the least competitive species. The average competitiveness rank is the mean of species' carbon and nitrogen ranks. Since the growth rate of a species limited by a resource c is given by λ^(c)^*c, with c – being its environmental concentration, the species with a higher λ^(c)^ outcompetes the species with a lower λ^(c)^ for the resource limiting to both of them.

The surprising result that a rank 5.5 species is uninvadable 20% of the time should be explained. What is a specific scenario in which this occurs, and why?

The factors deciding species prevalence are complex. In fact, only one of the 3 species with the mean competitiveness rank of 5.5 has high prevalence. It is decided by the interplay between competitiveness of all species in our pool. This particular species using c_1_ and n_1_ by chance was present as a ``satellite'' species in a state (or several states) with high prevalence. The goal of Figure 4B was to illustrate that the outcome of the competition in our model highly dependent on the composition of the entire species pool. In contrast, in a model with only one type of resource (say C), the species with the rank 1 would always win and all species with lower ranks would have zero prevalence.

6) Volume in abundance spaceThe authors focus on the feasibility volume of a state in the space of nutrients. However, it is also natural to consider the volume of the basins of attraction in the space of species abundances for a particular state in a given environment, when multiple stable states are possible. Does the volume of the basin of attraction for a state relate to its feasibility volume?

There is a close relationship between state's feasibility volume in the space of nutrients and its basin of attraction in the space of species abundances. They are identical up to the determinant of the matrix R_p_ in Equation 5 in Materials and methods. It could be understood using the specific example in Equation 6 describing the state #5 in our 2C x 2N x 4S model. To calculate the feasibility volume of this state we search for all vectors of ϕi (ϕi<ϕmax) for which it is feasible, that is to say, when all values of B and n in the left hand side of this equation are positive. To calculate the basin of attraction of species abundances, one goes in the opposite direction and scans all positive values of B_11_, B_12_, B_22_, and n_2_ between, say 0 and some upper bound, and constrain them by the condition ϕi<ϕmax for all nutrients. The elementary linear algebra dictates that the volumes of these two searches are connected via det(R_p_). In our manuscript we choose to use the feasibility volume since 1) it is more relevant to possible experiments, where it is easy to control nutrient supply rates but hard – bacterial abundances; 2) it has been previously studied in the literature (Grilli et al., 2016, Butler and O'Dwyer, 2018) and even given a special name "structural stability".